# CerS6-dependent ceramide synthesis in hypothalamic neurons promotes ER/mitochondrial stress and impairs glucose homeostasis in obese mice

Philipp Hammerschmidt[1,2,3], Sophie M. Steculorum [3,4,5], Cécile L. Bandet[1,2,3], Almudena Del Río-Martín [1,2,3], Lukas Steuernagel [1,2,3], Vivien Kohlhaas[1,2,3], Marvin Feldmann[1,2,3,6], Luis Varela[7,8,9], Adam Majcher[10,11], Marta Quatorze Correia[1], Rhena F. U. Klar[1], Corinna A. Bauder[1,2,3], Ecem Kaya[1,2,3], Marta Porniece [1,2,3], Nasim Biglari[1,2,3], Anna Sieben [1,2,3], Tamas L. Horvath [3,7,8,9], Thorsten Hornemann [10,11], Susanne Brodesser [3] & Jens C. Brüning [1,2,3,5] ✉

Dysregulation of hypothalamic ceramides has been associated with disrupted neuronal pathways in control of energy and glucose homeostasis. However, the specific ceramide species promoting neuronal lipotoxicity in obesity have remained obscure. Here, we find increased expression of the $C_{16:0}$ ceramide-producing ceramide synthase (CerS)6 in cultured hypothalamic neurons exposed to palmitate in vitro and in the hypothalamus of obese mice. Conditional deletion of *CerS6* in hypothalamic neurons attenuates high-fat diet (HFD)-dependent weight gain and improves glucose metabolism. Specifically, CerS6 deficiency in neurons expressing pro-opiomelanocortin (POMC) or steroidogenic factor 1 (SF-1) alters feeding behavior and alleviates the adverse metabolic effects of HFD feeding on insulin sensitivity and glucose tolerance. POMC-expressing cell-selective deletion of *CerS6* prevents the diet-induced alterations of mitochondrial morphology and improves cellular leptin sensitivity. Our experiments reveal functions of CerS6-derived ceramides in hypothalamic lipotoxicity, altered mitochondrial dynamics, and ER/mitochondrial stress in the deregulation of food intake and glucose metabolism in obesity.

Homeostatic control of energy metabolism relies on a central regulatory system involving specialized fuel-sensing neurons in the hypothalamus[1–3]. Important neurocircuits in control of energy balance are localized within the arcuate nucleus (ARC), containing orexigenic agouti-related peptide (AgRP)-expressing and anorexigenic pro-opiomelanocortin (POMC)-expressing cells that maintain a dynamic equilibrium in food intake and metabolic output[2,3]. Similarly, in the ventromedial nucleus of the hypothalamus (VMH), steroidogenic factor (SF)−1-expressing cells are associated with neuroendocrine functions related to glucose metabolism, feeding behavior, and thermogenesis[2,3]. These neuronal populations integrate hormonal and nutritional signals to modulate appetite, peripheral insulin sensitivity, glucose homeostasis, and energy expenditure, thus ensuring that the body's energy

needs are met and various metabolic processes remain in state of balance[2,3].

Sensing of fatty acids in the hypothalamus constitutes a pivotal mechanism for the regulation of diverse physiological responses, particularly under conditions of caloric excess or deprivation[4,5]. However, chronic overnutrition due to a high-fat diet (HFD) and the progression of obesity lead to excessive exposure of the brain to fatty acids, such as palmitate, along with their metabolites derived from local synthesis and circulation[6–10]. This condition, termed hypothalamic lipotoxicity, has detrimental effects on metabolic health. Lipotoxicity triggers metaflammation, endoplasmic reticulum (ER) stress, and mitochondrial dysfunction in the hypothalamus, contributing to local insulin and leptin resistance, disturbed neuronal activity, and disruptions in systemic energy and glucose regulation[11]. Nonetheless, the specific lipids and endogenous pathways responsible for hypothalamic lipotoxicity remain poorly defined. In light of the rising prevalence in obesity-related disorders, such as type 2 diabetes mellitus and cardiovascular disease, it has become necessary to identify the lipid species that contribute to hypothalamic dysfunction[12]. Advancements in this area could lead to more targeted and effective treatments, addressing the growing health challenges associated with obesity and related metabolic disorders.

Prior research has indicated that the consumption of HFD or the infusion of lipids into the periphery lead to accumulation of ceramides in the hypothalamus of rodents, and this has been correlated with metabolic deregulation[13]. Ceramides are a group of ubiquitously produced sphingolipids derived of a sphingoid base and a variety of fatty acids. They function as intracellular nutritional signals, playing a regulatory role in appetite control, fatty acid and glucose homeostasis, and the determination of cell fate[14]. Surplus of ceramides in peripheral tissues contributes to metabolic deregulation, while ceramide-lowering interventions in obese rodents have demonstrated the potential to improve insulin sensitivity and prevent the onset of future cardiometabolic diseases[15–19]. In the rodent hypothalamus, increased ceramide synthesis has been linked to ER stress, leading to reduced sympathetic outflow, diminished brown fat thermogenesis, and weight gain[20]. Conversely, pharmacological inhibition of central ceramide synthesis results in a reduction of ceramide levels in the hypothalamus of obese rats, enhancing hypothalamic insulin sensitivity and ameliorating systemic glucose tolerance[21].

Several species of ceramides exist, displaying divergent physicochemical and pathophysiological characteristics owing to variations in the composition of the sphingoid scaffold, acyl chain saturation, and acyl chain length[22]. The ceramide acyl chain length is determined by six (dihydro)ceramide synthases (CerS1-6), with each employing a restricted set of fatty acyl CoAs for N-acylation of long-chain bases, i.e., sphinganine during de novo synthesis, or sphingosine in the sphingolipid salvage pathway[23]. CerS1 has substrate preference for $C_{18}$ fatty acyl CoA, CerS2 for $C_{22}$-$C_{24}$, CerS3 for $C_{22}$-$C_{26}$, and CerS4 for $C_{18}$-$C_{20}$, while CerS5 and CerS6 possess overlapping specificities for $C_{14}$ and $C_{16}$ acyl CoAs[23]. These CerS enzymes display diverse expression profiles across cell types and organelles, resulting in a complex (sub)cellular distribution pattern of ceramides with different acyl chain lengths[24]. Previously, CerS6-derived $C_{16:0}$ ceramides and CerS1-derived $C_{18:0}$ ceramides have been identified as particularly detrimental in distinct peripheral tissues of obese rodents[25–30]. However, the specific acyl chain ceramide species contributing to hypothalamic lipotoxicity during the progression of obesity remained unclear.

In this work we investigated the potential role of specific CerSs and their associated ceramide products in hypothalamic neurons related to the deregulation of glucose and energy homeostasis in obesity. We analyzed cultured hypothalamic neurons exposed to palmitate in vitro and characterized mice fed a HFD with targeted CerS deficiencies in hypothalamic neurons. Our findings indicate that $C_{16:0}$ ceramides derived from CerS6 promote organellar stress and cause alterations in mitochondrial morphology and function within hypothalamic neurons, ultimately contributing to impairments of metabolic control linked to diet-induced obesity, particularly through actions in SF-1- and POMC- but not AgRP-expressing cells.

## Results

### CerS6 expression and $C_{16:0}$ ceramide levels are increased in the hypothalamus of obese mice

Ceramides have emerged as regulators of glucose- and energy homeostasis, partly through as of yet poorly defined effects in the central nervous system[31]. To investigate whether specific ceramide species are deregulated in the hypothalamus in obesity, we assessed the mRNA expression of the different CerSs by quantitative real-time PCR (qPCR)-based analysis in extracts of the hypothalamus derived from animal models of obesity and diabetes. To this end, we employed the genetically obese and hyperglycemic db/db mice and misty controls. In addition, we analyzed C57BL/6 N mice, in which obesity and insulin resistance had been induced by HFD feeding for 16 weeks or which received a low-fat control diet (CD; Supplementary Fig. 1a-c). Except for the lowly expressed CerS3, for which the detection threshold was not reached, all other CerS transcripts could be quantified in the hypothalamus of mice (Supplementary Fig. 1d). Specifically, we found increased mRNA expression of the $C_{16:0}$ ceramide-producing ceramide synthases CerS5 and CerS6 in the hypothalamus of both the db/db mouse model and HFD-fed mice compared to their respective lean counterparts, while the expression levels of CerS1, CerS2, and CerS4 were similar between the groups (Fig. 1a, b). The rise in CerS6 transcript levels in diet-induced obese animals translated into elevated hypothalamic CerS6 protein expression, as assessed by Western blot analysis in an additional cohort of HFD-fed mice (Fig. 1c). This was accompanied by an increase in the content of $C_{16:0}$ ceramides ( ~13%) and that of selected other ceramide, sphingomyelin, and hexosylceramide species in the hypothalamus of HFD-fed mice compared to CD-fed controls (Fig. 1d, Supplementary Fig. 1e, f). Indeed, of the two critical ceramide species, the levels of $C_{16:0}$ but not the more abundant $C_{18:0}$ ceramides in the hypothalamus showed a positive correlation with the animal's body weight (Fig. 1e, f).

Given the cellular heterogeneity in the hypothalamus, we analyzed the expression of the CerSs and their ceramide products across different hypothalamic cell types. We dissociated cells from the hypothalamus of regular chow-fed adult mice and isolated neuronal and non-neuronal cell fractions based on negative immunomagnetic selection of neurons and separation from non-neuronal cells (including astrocytes, oligodendrocytes, microglia, endothelial cells, and fibroblasts), followed by ceramide quantification in the two fractions. Successful fractionation of hypothalamic cell suspensions was verified by flow cytometry-based detection of CD11b (microglia), O4 (oligodendrocytes), and ACSA-2 (astrocytes) immunoreactivity (Supplementary Fig. 1g, h). Assessment of ceramide species indicated that the ceramide composition in hypothalamus lsyates closely resembles that found in isolated hypothalamic non-neuronal cells, with a $C_{18:0}$ ceramide-dominant phenotype (Fig. 1d, Supplementary Fig. 1i). Indeed, the neuron-enriched fraction showed a remarkably different pattern of ceramide levels compared to non-neuronal cells or the hypothalamus homogenate, with $C_{16:0}$ ceramides reflecting the predominant ceramide species in hypothalamic neurons (Fig. 1d, Supplementary Fig. 1i). To further reveal the cell-type-specific expression profiles of the individual CerS genes in the hypothalamus of mice, we determined the transcript levels of CerS1-6 by utilizing HypoMap, a harmonized transcriptomic reference map of the murine hypothalamus based on independent single-cell sequencing experiments[32]. Here, CerS1 and CerS3 were expressed at relatively low levels in all clusters of hypothalamic cells (Fig. 1g, h). Expression of CerS2, CerS4, and CerS5 was detectable across several different cell types, with CerS2 being

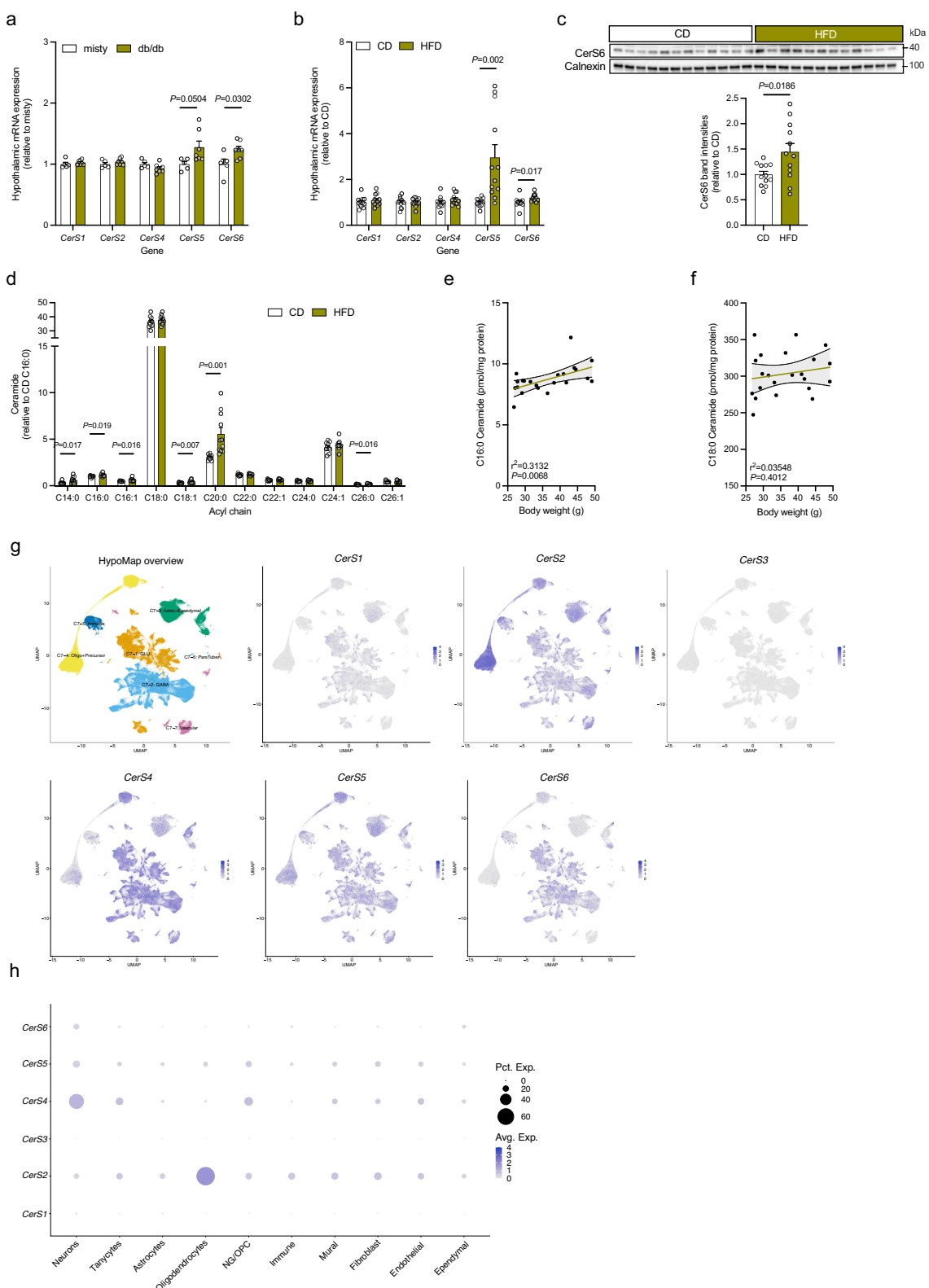

enriched in oligodendrocytes (Fig. 1g, h). Interestingly, *CerS6* transcripts were mainly found in the neuronal cluster (Fig. 1g, h), including SF-1-, POMC-, and AgRP-expressing cells (Supplementary Fig. 1j). Together, these observations suggest that the increase in $C_{16:0}$ ceramide levels and that of CerS6 expression in the hypothalamus of HFD-fed mice is mainly due to regulation in neurons compared to other cell types, indicative of a critical role of CerS6-derived $C_{16:0}$ ceramides in hypothalamic neurons in obesity.

### CerS6-derived ceramides promote palmitate-dependent ER/mitochondrial stress in cultured mouse hypothalamic neurons

We found here that (a) *CerS6* mRNA expression is enriched in neurons as compared to other cells of the hypothalamus, (b) $C_{16:0}$ ceramides are the predominant ceramide species in hypothalamic neuron-enriched cell fractions, and (c) hypothalamic CerS6 expression and $C_{16:0}$ ceramide content increase in obesity (see above). Furthermore, we have reported previously that body-wide ablation of CerS6- but not

**Fig. 1 | Hypothalamic CerS6 expression and $C_{16:0}$ ceramide levels are increased in mouse models of obesity and diabetes. a, b** Relative mRNA expression of the ceramide synthase (CerS) genes in hypothalamus samples of misty and db/db male mice (**a** $n = 5$ vs. 7 mice), and C57BL/6 N male mice fed a control diet (CD) or high-fat diet (HFD) for 16 weeks (**b** $n = 12$ vs. 12 mice). **c** Immunoblot and densitometric analysis of CerS6 in hypothalamus lysates of CD- and HFD-fed mice ($n = 12$ vs. 12 mice). CerS6 band intensities normalized to Calnexin; values expressed relative to CD. Uncropped blots in Source Data. **d** Ceramide levels in the hypothalamus of CD- and HFD-fed mice ($n = 11$ vs. 11 mice) relative to the $C_{16:0}$ ceramide content in CD-fed mice. **e, f** Linear regression model of the hypothalamic content of $C_{16:0}$ ceramide (**e**) and $C_{18:0}$ ceramide (**f**) plotted against the body weight of CD- and HFD-fed mice ($n = 22$ mice). Dotted lines indicate the 95% confidence interval; $r^2$ denotes the

coefficient of determination. **g** Uniform Manifold Approximation and Projection (UMAP) plots of single-cell sequencing data according to HypoMap. HypoMap overview is colored by major cell types, with the orange-colored cluster depicting GLUtamatergic neurons and the light blue-colored cluster depicting GABAergic neurons. Corresponding UMAP plots are colored by log-normalized gene expression of *CerS1-6*. **h** Dot plots displaying relative expression of *CerS1-6* in major cell types according to single-cell sequencing results harmonized in HypoMap. Data in **a**–**d** are represented as mean values ± SEM, including data points of individual mice entering the analysis. *P*-values calculated using two-tailed unpaired Student's *t*-test (**a**–**d**) or simple linear regression modeling (**e, f**). Source data and further details of statistical analyses are provided as a Source Data file.

CerS5-dependent ceramide synthesis protects mice from diet-induced obesity and insulin resistance[27]. Thus, we sought to study the specific role of CerS6-dependent ceramide formation in hypothalamic neurons in these processes. To this end, we employed the embryonic mouse (mHypoE)-N43/5 cell line, which is widely used as an in vitro system mimicking hypothalamic neuronal physiology[33,34]. We subjected these cells to fatty acid excess by incubation with palmitate, reflecting the environmental condition to which hypothalamic neurons are exposed to in obesity and metabolic syndrome[10,35]. Determination of *CerS* mRNA expression revealed a specific increase in *CerS6* transcripts following palmitate treatment in N43/5 cells (Fig. 2a), similar to what we observed in the hypothalamus of both diet-induced obese mice and db/db mice (see above). In turn, other *CerS* mRNAs remained unaffected, particularly that of *CerS5*, while *CerS3* was not detectable by qPCR analysis in N43/5 cells (Fig. 2a). Consistent with elevated *CerS6* expression, $C_{16:0}$ ceramide levels increased after palmitate exposure (Fig. 2b, Supplementary Fig. 2a). Importantly, the ceramide profile of cultured N43/5 cells resembled that of neurons enriched from hypothalamic cell suspensions (Supplementary Fig. 1i, 2a). These findings support the notion that $C_{16:0}$ ceramides may be of particular relevance in neurons of the hypothalamus upon fatty acid excess, and that N43/5 cells reasonably represent the ceramide distribution pattern of neurons in vivo, making them a suitable in vitro model to study the cellular effects of altered ceramide synthesis in hypothalamic neurons.

To investigate the functional role of CerS6 in hypothalamic neurons in particular, we performed knockdown experiments by siRNA-mediated RNA interference targeting the murine *CerS6* mRNA (siCerS6). siCerS6 treatment reduced *CerS6* mRNA expression in N43/5 cells approximately by half, which translated into reduced CerS6 protein levels by >80% (Fig. 2c, Supplementary Fig. 2b). In turn, *CerS4* transcript levels increased and that of other *CerSs* remained unchanged (Supplementary Fig. 2b). siCerS6 treatment fully abolished the palmitate-driven increase in *CerS6* mRNA expression (Fig. 2d), attenuating the accumulation of $C_{16:0}$ ceramides (Fig. 2e), which was accompanied by only a mild reduction in other selected ceramide species (Supplementary Fig. 2c). A similar effect was observed for the low abundant $C_{16:0}$ dihydroceramide and deoxyceramide species, indicating a general inhibition of the de novo synthesis pathway of $C_{16:0}$ ceramides upon knockdown of *CerS6* (Supplementary Fig. 2d, e). $C_{16:0}$ sphingomyelin levels did not increase in response to palmitate treatment and were only slightly decreased after *CerS6* knockdown (Supplementary Fig. 2f). $C_{16:0}$ hexosylceramides, in turn, accumulated in N43/5 cells upon knockdown of *CerS6*, possibly due to a compensatory upregulation of the sphingolipid salvage pathway, which did not sufficiently replenish cellular $C_{16:0}$ ceramide levels (Supplementary Fig. 2g). Indeed, $C_{16:0}$ ceramides were reduced up to 40% upon knockdown of *CerS6* compared to palmitate-treated control cells, 9 and 12 h after palmitate treatment (Fig. 2e).

Next, we aimed to elucidate the molecular consequence of inhibiting CerS6-dependent synthesis of $C_{16:0}$ ceramides in cultured hypothalamic neurons. Knockdown of *CerS6* in N43/5 cells attenuated the palmitate-dependent increase in splicing of the mRNA encoding X-

box-binding protein 1 (Xbp1; Fig. 2f). Spliced *Xbp1* (*sXbp1*) encodes a transcription factor that promotes the expression of proteins involved in the unfolded protein response of the ER ($UPR^{ER}$), and has been shown to connect ER stress in hypothalamic neurons with energy balance and glucose homeostasis[36]. In line with reduced *Xbp1* splicing, the depletion of CerS6 attenuated the palmitate-induced upregulation of transcripts for the ER stress marker genes *Ddit3*, which encodes the ER stress-induced C/EBP homologous protein (CHOP), and *Hspa5*, which encodes the ER luminal chaperone BiP/GRP78 (Fig. 2g). We also tested for transcript levels of markers for mitochondrial stress. These included genes encoding (a) the activating transcription factor ATF4, which plays a critical role in the initiation of both $UPR^{ER}$ and the mitochondrial UPR ($UPR^{mt}$)[37], (b) the fibroblast growth factor FGF21, which is a proteotypic effector of the mitochondrial damage stress response also in neurons[38], (c) the growth/differentiation factor GDF15, which provides an endocrine signal of nutritional and mitochondrial dysfunction[39], and (d) the mitochondrial protease ClpP, which degrades misfolded proteins during $UPR^{mt}$ (Fig. 2h; from left to right). Interestingly, palmitate upregulated the expression of these mitochondrial stress-associated genes in control cells, a response that was mitigated upon knockdown of *CerS6* (Fig. 2h). The role of CerS6-dependent ceramide synthesis in ER/mitochondrial stress regulation was supported by the reduction in protein levels of BiP/GRP78 and ATF4 in cells treated with siCerS6, as compared to control cells, under conditions of palmitate exposure (Fig. 2i). Notably, palmitate not only induced ER/mitochondrial stress in N43/5 cells but also led to a significant decrease in mean mitochondrial aspect ratio, indicating a more fragmented mitochondrial network, as assessed by electron microscopy-based imaging of palmitate-treated cells (Fig. 2j, k). Importantly, the mitochondrial fragmentation resulting from palmitate exposure was alleviated in cultured hypothalamic neurons upon the depletion of CerS6-dependent ceramide formation (Fig. 2j, k), concomitant with higher mitochondrial respiratory activity (Fig. 2l).

These data indicate that palmitate stimulates CerS6-dependent $C_{16:0}$ ceramide synthesis, which promotes ER/mitochondrial stress and mitochondrial fragmentation in hypothalamic neurons, in conjunction with reduced mitochondrial bioenergetic activity; this may contribute to the alterations in ER/mitochondrial plasticity and function observed in neurons of the hypothalamus in vivo in response to fatty acid excess, leading to adverse metabolic consequences in obesity[40].

## CerS6 in hypothalamic neurons promotes HFD-induced obesity and insulin resistance in mice

Palmitate excess in cultured hypothalamic neurons induces CerS6-dependent ceramide formation, which promotes fatty acid-induced ER/mitochondrial stress and affects mitochondrial dynamics with alterations in bioenergetic efficacy (see above). These processes have previously been linked to changes in neuronal physiology that contribute to impairments in glucose metabolism[40–42]. Thus, we aimed to investigate the specific role of hypothalamic CerS6-dependent ceramide synthesis during obesity progression in vivo. As it is unknown whether CerS6 deficiency would affect the $C_{16:0}$ ceramide content in the

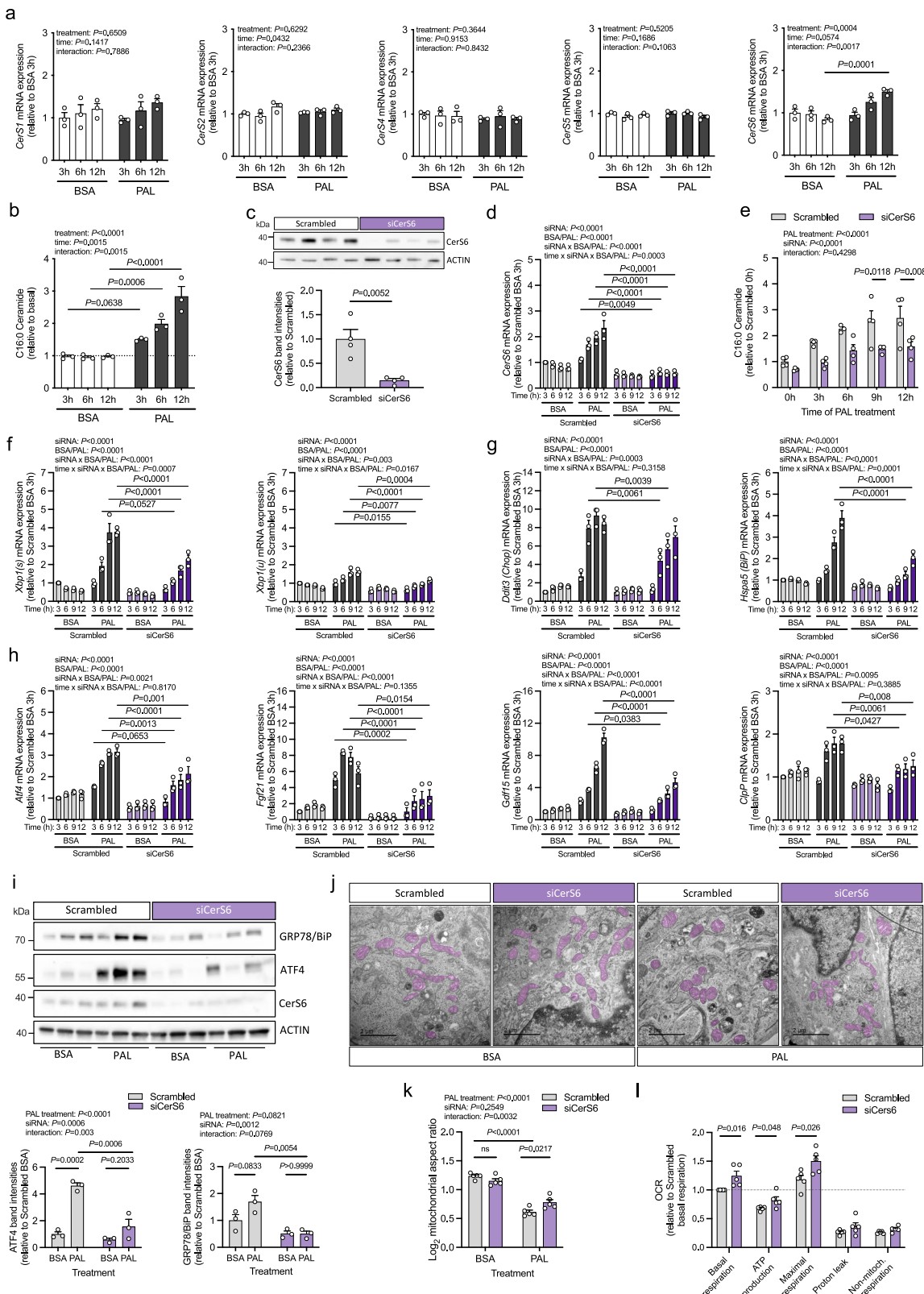

hypothalamus upon obesogenic dieting, we first performed lipidomic analysis in hypothalamus lysates of HFD-fed control mice and conventional CerS6 knockouts (*CerS6*$^{\Delta/\Delta}$). *CerS6*$^{\Delta/\Delta}$ mice have been generated previously by Cre/loxP-mediated germline deletion of exon 4 in *CerS6*, which produces a frameshift in the downstream exons 5-8 to prevent translation of the catalytic domain, thus inhibiting CerS6-dependent ceramide synthesis throughout the offspring's body (Supplementary

Fig. 3a)[25]. These animals lack CerS6 protein also in the hypothalamus (Supplementary Fig. 3b) and, as reported in more detail previously, are protected from diet-induced obesity and the deterioration of glucose metabolism (Supplementary Fig. 3c)[25]. Indeed, HFD-fed CerS6-deficient mice exhibit reduced levels of C$_{16:0}$ ceramides and a slight increase in C$_{20:0}$ ceramides in the hypothalamus compared to HFD-fed obese controls (Supplementary Fig. 3d). In contrast, C$_{16:0}$ sphingomyelin and

**Fig. 2 | Inhibition of CerS6-dependent ceramide synthesis in hypothalamic neurons attenuates palmitate-induced ER/mitochondrial stress and mitochondrial fragmentation. a** Relative mRNA expression of *CerS1-6* in N43/5 cells following palmitate (PAL, black) or BSA treatment (BSA, white) for the time indicated in hours (h) (*n* = 3 independent experiments). **b** C$_{16:0}$ ceramide levels relative to the C$_{16:0}$ ceramide content in untreated cells (basal, dotted line) (*n* = 3 independent experiments). **c** Immunoblot analysis of CerS6, 57 h after transfection with an siRNA targeting *CerS6* (siCerS6, purple) or scrambled control (Scrambled, gray). CerS6 band intensities normalized to ACTIN; values expressed as relative to control (*n* = 4 independent experiments). Uncropped blots in Source Data. **d** Relative mRNA expression of *CerS6* in siRNA-treated cells incubated with BSA or PAL for the time indicated (*n* = 3 independent experiments). **e** Relative C$_{16:0}$ ceramide levels in siRNA-treated cells after incubation with PAL for the time indicated (*n* = 4 independent experiments; *n* = 3 for Scrambled 6 h). **f–h** Relative mRNA expression of spliced *Xbp1* (*Xbp1(s)*) and unspliced *Xbp1* (*Xbp1(u)*) (**f**), the ER stress markers *Ddit3* and *Hspa5* (**g**), and the mitochondrial stress markers *Atf4, Fgf21, Gdf15,* and *ClpP* (**h**) in siRNA-treated cells incubated with BSA or PAL for the time indicated (*n* = 3

independent experiments). **i** Immunoblot analysis of ATF4 (left quantification), GRP78/BiP (right quantification), and CerS6 in siRNA-treated cells incubated with BSA or PAL for 9 h. Band intensities normalized to ACTIN; values expressed as relative to BSA-treated scrambled control (*n* = 3 independent cultures). Uncropped blots in Source Data. **j** TEM images of siRNA-treated cells incubated with BSA or PAL for 9 h (mitochondria are highlighted in purple, scale bars: 2 μm). **k** Quantification of log$_2$-transformed mitochondrial aspect ratios from TEM images as represented in (**j**) (*n* = 5 independent experiments; aspect ratios from a minimum of 10 cells/group/experiment). **l** Mitochondrial oxygen consumption rates (OCR) in siRNA-treated N43/5 cells incubated with PAL for 9 h (*n* = 5 independent experiments with 46 technical replicates per run). Data represented as mean values of n independent experiments ±SEM. Individual data points in (**d, f–h**) represent the average of three replicate culture dishes per experiment. *P*-values calculated using two-tailed unpaired Student's *t*-test (**c, l**), two-way ANOVA and Bonferroni's (**a, b, e**) or Tukey's multiple comparison test (**i, k**), or three-way ANOVA and Tukey's multiple comparison test (**d, f–h**). Source data and further details of statistical analyses are provided as a Source Data file.

hexosylceramide levels remained unaltered in *CerS6*$^{Δ/Δ}$ mice, while C$_{18:0}$ and C$_{20:0}$ sphingomyelin levels increased, signifying a selectivity for a reduction in C$_{16:0}$ ceramides upon inhibiting CerS6-dependent ceramide synthesis within the hypothalamus (Supplementary Fig. 3e, f).

We then aimed to target CerS6 specifically in hypothalamic neurons. Mice, in which exon 4 of *CerS6* was flanked by loxP sites (*CerS6*$^{fl/fl}$), were bred to those expressing the Cre recombinase under the control of the Nkx2.1 promoter predominantly in a large proportion of hypothalamic neurons (Fig. 3a)[43,44]. This yielded Nkx2.1-Cre-positive *CerS6*$^{fl/fl}$ mice (*CerS6*$^{ΔNkx2.1}$) and Cre-negative *CerS6*$^{fl/fl}$ littermates (Control). Successful genetic recombination of the floxed *CerS6* allele was verified by PCR of genomic DNA extracted from the hypothalamus, while no Cre-mediated recombination was detected in the tail of *CerS6*$^{ΔNkx2.1}$ mice or in control animals (Supplementary Fig. 3g). Further validation of the mouse model was accomplished by the assessment of *CerS6* mRNA expression through in situ hybridization on brain slices. To this end, specific BaseScope probes were used targeting either the exon junction 4/5 of the *CerS6* transcript (*CerS6*$^{E4E5}$) to assess expression of full-length *CerS6* containing exon 4, or the exon junction 3/5 (*CerS6*$^{E3E5}$) to detect the exon 4-deficient modified transcript of *CerS6* (Supplementary Fig. 3h). In control mice, full-length *CerS6* transcripts were found at different sites of the hypothalamus, including the ARC, VMH, and dorsomedial nucleus (DMH), as well as other brain regions such as the dentate gyrus (DG) of the hippocampus (Supplementary Fig. 3h). In comparison, in the hypothalamus of *CerS6*$^{ΔNkx2.1}$ mice, full-length *CerS6* was clearly reduced (Supplementary Fig. 3h). In turn, *CerS6*$^{ΔNkx2.1}$ mice showed expression of the exon 4-deficient transcript of *CerS6* most prominently in cells of the hypothalamus without significant off-target detection in the hippocampus or in control animals (Supplementary Fig. 3h). qPCR-based mRNA expression analysis using a probe targeting exon 4 in the *CerS6* cDNA confirmed the reduction of *CerS6* expression specifically in the hypothalamus of *CerS6*$^{ΔNkx2.1}$ mice (Fig. 3b). *CerS6* transcript levels remained similar between *CerS6*$^{ΔNkx2.1}$ mice and controls in the rest of the brain, liver, and gonadal white adipose tissue (gWAT) (Supplementary Fig. 3i, j). At the same time, expression of *CerS1* and *CerS5* remained unchanged in the hypothalamus and rest brain, indicative of successful and specific modification of hypothalamic *CerS6* expression in *CerS6*$^{ΔNkx2.1}$ mice (Fig. 3b, Supplementary Fig. 3i). In accordance with decreased mRNA expression, CerS6 protein levels were markedly reduced in the hypothalamus of *CerS6*$^{ΔNkx2.1}$ mice compared to controls (Fig. 3c).

For metabolic characterization during the development of diet-induced obesity, *CerS6*$^{ΔNkx2.1}$ mice and their control littermates were subjected to HFD feeding starting at 3 weeks of age, followed by the analysis of body weight gain, adiposity, energy balance, and glucose metabolism. Deletion of *CerS6* in Nkx2.1-expressing cells attenuated weight gain of mice exposed to the HFD, while it did not affect body

length as measured in adult animals (Fig. 3d, e). Analysis of body composition revealed reduced adiposity in HFD-fed *CerS6*$^{ΔNkx2.1}$ mice compared to controls (Fig. 3f). Consistent with lower fat mass, leptin levels were decreased in serum samples of random-fed *CerS6*$^{ΔNkx2.1}$ animals (Fig. 3g). Notably, the reduction in adiposity was likely due to increased energy expenditure in *CerS6*$^{ΔNkx2.1}$ mice when compared to controls, especially considering that food intake remained similar between the two groups (Fig. 3h–j). In addition, *CerS6*$^{ΔNkx2.1}$ mice exhibited improved insulin sensitivity and an improved ability to lower blood glucose levels after intraperitoneal administration of glucose during a glucose tolerance test, particularly following an overnight fasting period (Fig. 3k–m). These metabolic enhancements were limited to mice fed the HFD, whereas normal chow diet (NCD)-fed *CerS6*$^{ΔNkx2.1}$ mice did not display changes in insulin sensitivity and only a tendency toward improved glucose tolerance compared to lean control mice (Supplementary Fig. 3k, l). These data imply that genetic inhibition of CerS6-dependent ceramide synthesis in hypothalamic neurons of mice attenuates both diet-induced adiposity and the obesity-associated impairments of glucose metabolism.

**CerS1 in hypothalamic neurons does not affect glucose metabolism upon HFD feeding in mice**

In addition to our observation that CerS6-derived C$_{16:0}$ ceramides promote metabolic deterioration in obesity[25,27], we have previously identified a role of skeletal muscle CerS1 and its product C$_{18:0}$ ceramide in the development of obesity-associated insulin resistance[25]. While both conventional and skeletal muscle-specific deletion of *CerS1* improve insulin sensitivity in HFD-fed mice, only the former approach prevents diet-induced obesity but at the expense of cerebellar hypoplasia[29]. In addition, previous studies indicate that CerS1 expression is most abundant in muscle and brain, in conjunction with C$_{18:0}$ ceramides being the predominant ceramide species in muscle and brain extracts[45,46]. Although our data suggest that *CerS1* is barely expressed in the hypothalamus and that most of the hypothalamic C$_{18:0}$ ceramide content originates from non-neuronal cells (see above), it cannot be excluded that they play a substantial role in hypothalamic neurons in vivo. In particular, we found that C$_{18:0}$ ceramides account for a major proportion of the total ceramide content also in neuron-enriched cell fractions (see above). Thus, we sought to investigate whether hypothalamic CerS1 plays a role in glucose metabolism similar to what the above-described experiments had revealed for CerS6, by targeting *CerS1* specifically in neurons of the hypothalamus. We generated mice with deficiency for *CerS1* in Nkx2.1-expressing cells (*CerS1*$^{ΔNkx2.1}$) by crossing *CerS1*$^{fl/fl}$ mice, in which exon 2 of *CerS1* was flanked by loxP sites, with those expressing the Nkx2.1-Cre transgene (Supplementary Fig. 4a)[29]. PCR analysis of genomic DNA confirmed the Cre-driven recombination of *CerS1* in the hypothalamus of *CerS1*$^{ΔNkx2.1}$

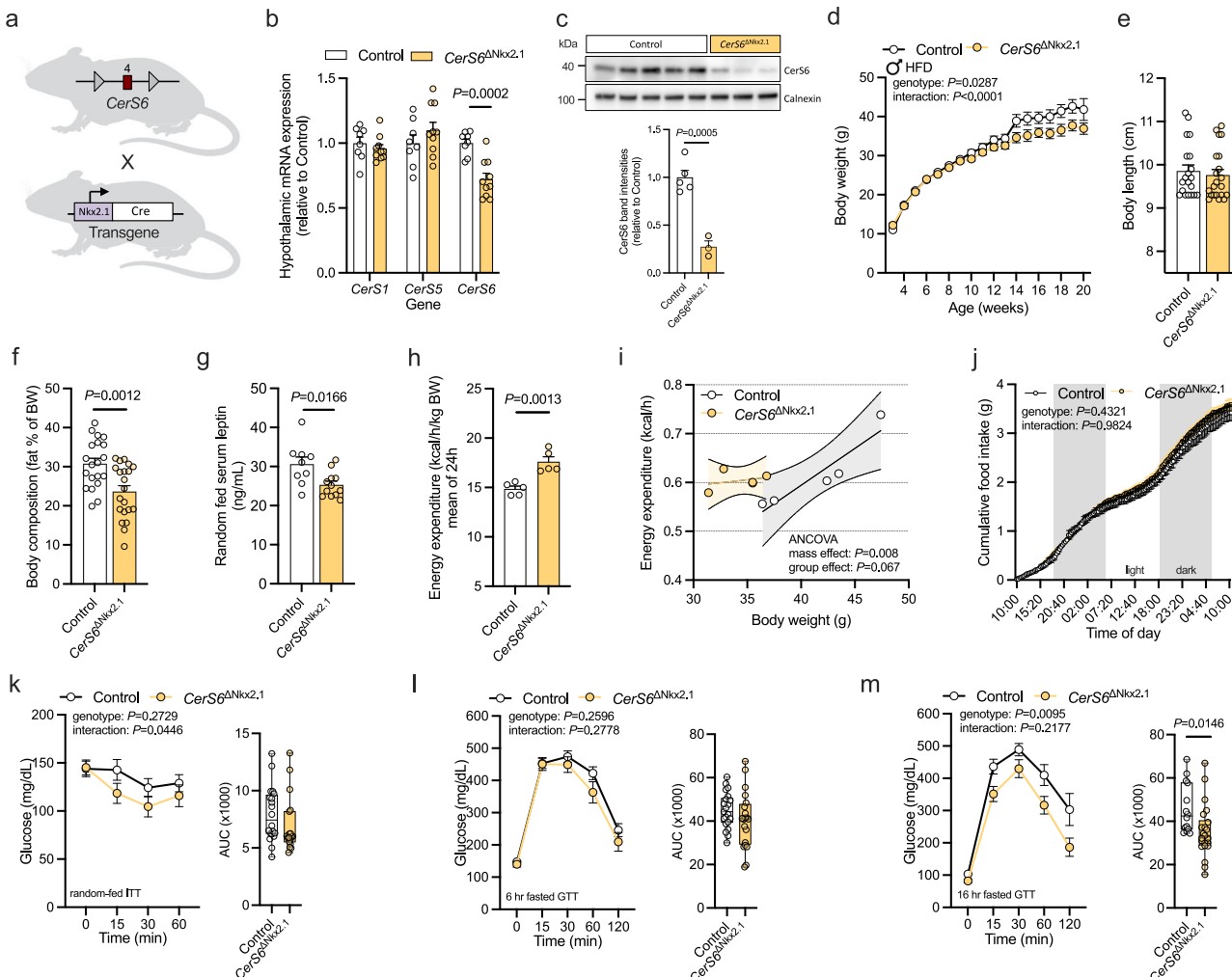

**Fig. 3 | Depletion of CerS6 in hypothalamic neurons attenuates diet-induced obesity and improves insulin sensitivity and energy metabolism in HFD-fed male mice. a** *CerS6*[fl/fl] mice in which exon 4 of *CerS6* (red square) is flanked by loxP sites (gray triangles) were bred to mice with transgenic expression of the Cre recombinase (white rectangle) under control of the Nkx2.1 promoter (purple rectangle). Nkx2.1-Cre-positive *CerS6*[fl/fl] mice (*CerS6*[ΔNkx2.1], yellow) and Cre-negative *CerS6*[fl/fl] control littermates (Control, white) were used for analysis. **b** Relative mRNA expression of *CerS1*, *CerS5*, and *CerS6* in hypothalamus samples of HFD-fed control animals and *CerS6*[ΔNkx2.1] mice determined by qPCR analysis (*n* = 8 vs. 10 mice). **c** Immunoblot analysis of CerS6 in hypothalamus lysates of controls and *CerS6*[ΔNkx2.1] mice fed a HFD (*n* = 5 vs. 3 mice). CerS6 protein band intensities normalized to Calnexin and values expressed as relative to control. Uncropped blots in Source Data. **d** Body weight development of mice during HFD feeding (*n* = 7–35 vs. 11–36 mice/week). **e** Body length in adult mice (*n* = 20 vs. 21 mice). **f** Body fat content

relative to body weight (*n* = 20 vs. 21 mice). **g** Random-fed serum leptin levels (*n* = 9 vs. 12 mice). **h** Average energy expenditure over 24 h normalized for body weight (*n* = 5 vs. 5 mice). **i** Regression-based analysis of absolute energy expenditure against body weight (*n* = 5 vs. 5 mice). Dotted lines indicate the 95% confidence interval. **j** Cumulative food intake over 48 h (*n* = 5 vs. 6 mice; gray background indicates dark phase). **k** Insulin tolerance test and area under the curve (AUC) for each mouse (*n* = 20 vs. 18 mice). **l** Glucose tolerance test following a 6 h fasting period and AUCs (*n* = 21 vs. 20 mice). **m** Glucose tolerance test following a 16 h fasting period and AUCs (*n* = 14 vs. 22 mice). All data obtained from male mice. Data in (**b**–**h**) and longitudinal data in (**j**–**m**) are represented as mean values ± SEM. *P*-values calculated using two-tailed unpaired Student's *t*-test (**b, c, f–h**, and AUC in **m**), a mixed-effects model (**d**), analysis of covariance (ANCOVA) (**i**), or two-way RM ANOVA (**j**, longitudinal analysis in **k–m**). Source data and further details of statistical analyses are provided as a Source Data file. Illustration in (**a**) created with BioRender.com.

mice (Supplementary Fig. 4b). In situ hybridization using a probe targeting exon junction 2/3 of the *CerS1* transcript (*CerS1*[E2E3]) revealed that full-length *CerS1* is expressed in the hypothalamus and in other brain regions such as the cerebral cortex of control animals, but is almost fully abolished in the hypothalamus of *CerS1*[ΔNkx2.1] mice (Supplementary Fig. 4c). Instead, *CerS1*[ΔNkx2.1] mice showed expression of the recombined exon 2-deleted transcript of *CerS1* in the hypothalamus, as assessed by using a probe specific for exon junction 1/3 in *CerS1*, indicating successful targeting of CerS1 in the hypothalamus of *CerS1*[ΔNkx2.1] mice (Supplementary Fig. 4c).

In contrast to what we observed in mice with *CerS6* deficiency in Nkx2.1-expressing cells, inhibiting CerS1 in neurons of the hypothalamus did not affect HFD-induced obesity, systemic insulin sensitivity, or glucose tolerance (Supplementary Fig. 4d-i). Indeed, *CerS1*[ΔNkx2.1] mice

showed similar degrees of weight gain, adiposity, insulin resistance, and glucose intolerance upon HFD feeding as their control littermates (Supplementary Fig. 4d-i). These findings highlight the cell-type-specific roles of the two different CerSs and their ceramide products in the development of obesity-related metabolic disorders. Hence, these data indicate that CerS1-derived $C_{18:0}$ ceramides in neurons of the hypothalamus do not promote the obesity-associated deterioration of glucose homeostasis, as opposed to their critical role in skeletal muscle and that of hypothalamic CerS6.

## CerS6 in SF-1-expressing neurons of the VMH affects energy and glucose metabolism in HFD-fed mice

*CerS6* but not *CerS1* deficiency in hypothalamic neurons alleviates diet-induced obesity and defective glucose handling (see above). We next

aimed to identify the specific hypothalamic neuronal populations in which CerS6-derived ceramides impinge on metabolic control. Given the critical role of the VMH in this process and the expression of *CerS6* in VMH neurons (see above), we sought to investigate whether CerS6-dependent $C_{16:0}$ ceramide synthesis would play a role in metabolically relevant neurons of the VMH. To this end, *CerS6*^fl/fl mice were bred to mice with transgenic expression of the Cre recombinase specifically under the control of the SF-1 promoter (Fig. 4a). We performed PCR analysis of genomic DNA extracted from ARC- and VMH-containing tissue samples and the rest of the hypothalamus of SF-1-Cre-positive *CerS6*^fl/fl mice (*CerS6*^ΔSF-1) and Cre-negative *CerS6*^fl/fl littermates, depicting the deletion of exon 4 in *CerS6* only in the VMH of *CerS6*^ΔSF-1 animals (Supplementary Fig. 5a). In line, BaseScope duplex in situ hybridization for *Nr5a1* (encoding SF-1) and *CerS6* revealed prominent expression of full-length *CerS6* in the VMH of control animals and a close association of *CerS6* with *Nr5a1*-positive cells, which was considerably reduced in

the VMH of *CerS6*^ΔSF-1 mice (Supplementary Fig. 5b). Conversely, the exon 4-deficient transcript modification of *CerS6* was detectable specifically in *Nr5a1*-positive cells in the VMH of *CerS6*^ΔSF-1 mice but not in control animals (Supplementary Fig. 5b).

Having validated the successful ablation of *CerS6* expression in SF-1 neurons of *CerS6*^ΔSF-1 mice, we subjected these animals and their control littermates to HFD feeding. We did not observe significant differences in weight gain and adiposity between *CerS6*^ΔSF-1 male mice and controls (Fig. 4b-e). In turn, obesity development was attenuated in *CerS6*^ΔSF-1 female mice, as evidenced by reduced weight gain and body fat content, weights of gWAT and subcutaneous WAT (sWAT) depots, and circulating leptin levels compared to their controls (Supplementary Fig. 5c-f). Since ceramide accumulation in the hypothalamus has been linked to ER stress in the VMH, diminishing BAT thermogenesis and energy expenditure[20], we performed indirect calorimetry measurements to assess the metabolic rate in *CerS6*^ΔSF-1

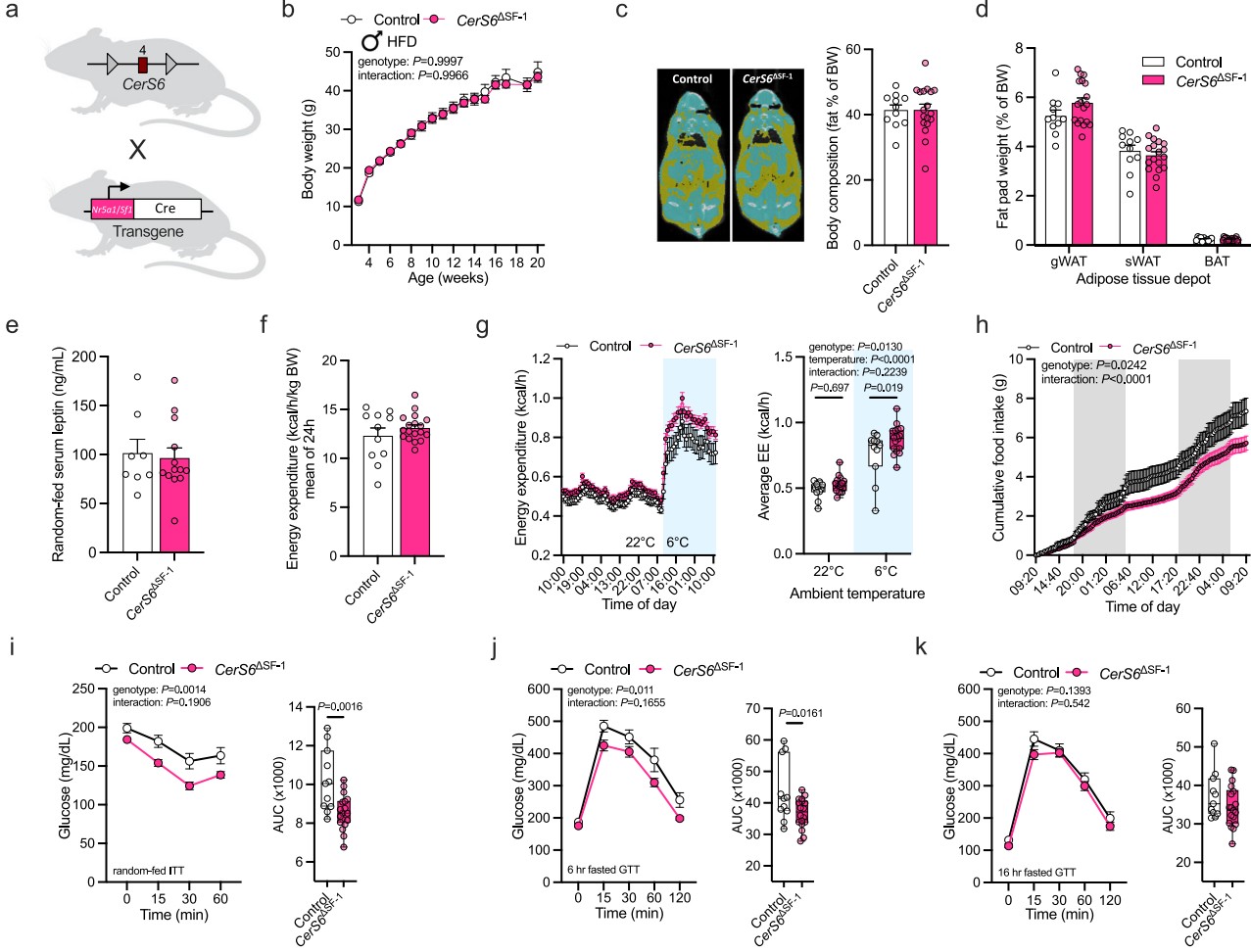

**Fig. 4 | Depletion of CerS6 in SF-1 neurons affects cold-induced energy expenditure and improves glucose metabolism in HFD-fed male mice. a** *CerS6*^fl/fl mice in which exon 4 of *CerS6* (red square) is flanked by loxP sites (gray triangles) were bred to mice carrying a transgene for Cre expression (white rectangle) under control of the SF−1 promoter (pink rectangle). SF-1-Cre-positive *CerS6*^fl/fl mice (*CerS6*^ΔSF-1, pink) and Cre-negative *CerS6*^fl/fl littermate controls (Control, white) were used for analysis. **b** Body weight development of mice during HFD feeding (*n* = 6–11 vs. 9–18 mice/weeks). **c** CT scans (yellow, fat tissue; blue, non-adipose soft tissue) and body fat content relative to body weight (*n* = 11 vs. 18 mice). **d** Fat-pad weights of gonadal white adipose tissue (gWAT), subcutaneous WAT (sWAT), and brown adipose tissue (BAT) relative to body weight (*n* = 11 vs. 18 mice). **e** Serum leptin levels of random-fed mice (*n* = 8 vs. 13 mice). **f** Average energy expenditure over 24 h normalized for body weight (*n* = 11 vs. 18 mice). **g** Absolute energy expenditure

(EE) at 22 °C ambient temperature (white background) and upon exposure to 6 °C (blue background), over time (left) and in average (right; *n* = 11 vs. 18 mice). **h** Cumulative food intake over 48 h (*n* = 11 vs. 18 mice, gray background indicates dark phase). **i** Insulin tolerance test and area under the curve (AUC) for each mouse (*n* = 11 vs. 18 mice). **j, k** Glucose tolerance test following a 6 h (**j**) or 16 h (**k**) fasting period and AUCs (*n* = 11 vs. 18 mice). All data obtained from male mice. Data in (**b**–**f**) and longitudinal data in (**g**–**k**) are represented as mean values ± SEM. Boxplots indicate median ±min/max and include data points of individual mice entering the analysis. *P*-values calculated using two-tailed unpaired Student's *t*-test (AUCs in **i, j**), two-way ANOVA and Bonferroni's multiple comparison test (Average EE in **g**), two-way RM ANOVA (longitudinal analysis in **h**–**k**), or a mixed-effects model (**b**). Source data and further details of statistical analyses are provided as a Source Data file. Illustration in (**a**) created with BioRender.com.

mice. Female and male *CerS6*[ΔSF-1] mice fed a HFD displayed a modest tendency toward increased energy expenditure at ambient temperatures, with statistically significant increases observed in male *CerS6*[ΔSF-1] mice compared to controls during exposure to a temperature of 6 °C (Fig. 4f, g, Supplementary Fig. 5g, h). In addition, food intake was reduced in *CerS6*[ΔSF-1] males but not females compared to their respective control animals (Fig. 4h, Supplementary Fig. 5i). Although female mice with CerS6 deficiency in SF-1 neurons exhibited a reduction in adiposity, they did not demonstrate substantial improvements in glucose metabolism (Supplementary Fig. 5j-l). In contrast, male *CerS6*[ΔSF-1] mice presented with significantly enhanced insulin sensitivity and glucose tolerance (Fig. 4i-k).

Of note, the SF-1 promoter is also active in the adrenals, and Cre activity in SF-1-Cre-expressing mice has been detected in the adrenal cortex, which executes important functions in maintaining metabolic integrity by secreting corticosteroids. To determine whether this process would be affected by deleting *CerS6* in SF-1-expressing cells, we measured serum corticosterone levels in HFD-fed male and female mice, which were not altered in *CerS6*[ΔSF-1] animals (Supplementary Fig. 5m, n).

Collectively, CerS6-dependent ceramide synthesis in SF-1-expressing neurons of the VMH controls glucose metabolism alongside minor effects on energy expenditure and food consumption in male mice and adiposity in females.

## CerS6 in AgRP neurons does not affect glucose metabolism in HFD-fed mice

Ablation of *CerS6* expression either in a large proportion of hypothalamic neurons or selectively in SF-1-expressing cells of the VMH improves metabolic homeostasis in obese mice (see above). Given the profound roles of specific neuronal populations located in the ARC of the hypothalamus on feeding behavior and glycemic control, we sought to investigate whether CerS6 additionally acts in specific ARC neurons to control appetite and glucose metabolism. Thus, we targeted CerS6-dependent ceramide synthesis specifically in AgRP- and POMC-expressing cells, which are recognized as crucial nodes in energy homeostasis with largely antagonistic functions[47].

First, *CerS6*[fl/fl] mice were bred to those expressing Cre from the endogenous AgRP locus (AgRP-IRES-Cre) to restrict the inhibition of CerS6-dependent ceramide synthesis to AgRP-expressing cells (*CerS6*[ΔAgRP], Fig. 5a). PCR analysis of genomic DNA confirmed the recombination of *CerS6* in the ARC of *CerS6*[ΔAgRP] mice, while no such recombination was observed in the rest of the hypothalamus of *CerS6*[ΔAgRP] mice or in controls (Supplementary Fig. 6a). In situ hybridization revealed close association between the full-length transcript of *CerS6* and *Agrp* in ARC cells of control animals, which was diminished in *CerS6*[ΔAgRP] mice (Supplementary Fig. 6b). The detection of the exon 4-deficient *CerS6* transcript modification was limited to *Agrp*-positive cells in the ARC of *CerS6*[ΔAgRP] mice, with no corresponding signal observed in other hypothalamic nuclei, such as the VMH, in both *CerS6*[ΔAgRP] mice and controls animals (Supplementary Fig. 6b). These results indicate the effective and targeted modification of CerS6 within AgRP neurons in this mouse model.

The specific deletion of *CerS6* in AgRP neurons of mice did not impact body weight gain or adiposity in either males or females when subjected to the HFD, with only a mild reduction in the relative wet weight of the anterior sWAT depot in male animals (Fig. 5b–d, Supplementary Fig. 6c–e). Consistently, the levels of circulating leptin were comparable between HFD-fed *CerS6*[ΔAgRP] mice and controls, as were the rates of energy expenditure and food intake (Fig. 5e–g, Supplementary Fig. 5f–h). Additionally, the evaluation of glucose metabolism demonstrated similar levels of insulin sensitivity and glucose tolerance between HFD-fed *CerS6*[ΔAgRP] mice of both sexes and their respective control littermates (Fig. 5h–j, Supplementary Fig. 6i–k). These observations indicate that CerS6 in AgRP neurons

does not contribute to the deterioration of energy and glucose homeostasis in obesity.

## CerS6 in POMC neurons affects mitochondrial morphology, leptin sensitivity, feeding behavior, and glucose metabolism in HFD-fed mice

In contrast to the adjacent AgRP neurons, the activity of POMC neurons in the ARC increases in response to food consumption, particularly after refeeding following a period of starvation or even shortly before eating, thereby anticipating satiety and priming metabolic liver adaptation to the postprandial state[48,49]. Thus, POMC neurons coordinate meal size and at the same time modulate peripheral glucose metabolism[50]. Notably, the regulation of mitochondrial morphology and leptin signaling in POMC neurons has been linked to the control of energy balance and body weight homeostasis[40]. POMC neuron activity is significantly reduced in obese mice, and experimental depletion of POMC neurons is sufficient to induce hyperphagia and obesity[51–53]. Therefore, we aimed to investigate whether CerS6 in POMC neurons plays a role in these processes in mice fed a HFD.

To generate mice with deficiency for CerS6-dependent ceramide formation specifically in POMC neurons (*CerS6*[ΔPOMC]), *CerS6*[fl/fl] mice were bred to those carrying a transgene for expression of Cre driven by the POMC promoter (Fig. 6a). As before, we verified the recombination of *CerS6* by PCR analysis of genomic DNA in ARC-containing tissue samples, while no recombination was found in the rest of the hypothalamus of *CerS6*[ΔPOMC] mice or in control animals (Supplementary Fig. 7a). In situ hybridization confirmed the expression of full-length *CerS6* transcripts in *Pomc*-positive cells in the ARC of control mice, which was decreased in *Pomc*-positive cells of *CerS6*[ΔPOMC] mice (Supplementary Fig. 7b). Conversely, the exon 4-deficient *CerS6* transcript was primarily found in cells co-expressing *Pomc*, and in a few *Pomc*-negative cells, within *CerS6*[ΔPOMC] mice, while it was not detected in the hypothalamus of control animals (Supplementary Fig. 7b). Of note, the recombination of *CerS6* in a minor fraction of non-POMC cells in *CerS6*[ΔPOMC] mice is probably attributed to the temporary activity of the *Pomc* promoter during embryonic development in hypothalamic cells that subsequently undergo a fate transition to non-POMC identities, i.e., AgRP neurons in the ARC, as has been previously demonstrated[54].

*CerS6*[ΔPOMC] mice of both sexes did not exhibit significant changes in body weight gain or adiposity when exposed to the HFD, in conjunction with no alterations in basal energy expenditure (Fig. 6b–f, Supplementary Fig. 7c–f). However, while basal food intake was not different between groups, specifically among fasted male *CerS6*[ΔPOMC] mice, a reduction in food intake was observed in comparison to controls animals during the dark phase following refeeding, a period when POMC neuron activity is expected to be increased (Fig. 6g, h, Supplementary Fig. 7g, h). Moreover, deficiency for *CerS6* in POMC neurons led to improved glucose metabolism in obese male mice but did not yield the same effect in females (Fig. 6i–k, Supplementary Fig. 7i–k). Specifically, male *CerS6*[ΔPOMC] mice fed the HFD displayed improvements in insulin sensitivity and glucose tolerance compared to their obese counterparts (Fig. 6i–k).

Next, we evaluated mitochondrial morphology in POMC neurons of these mice using electron microscopy. Our analysis revealed that feeding of a HFD for 8 weeks is linked to a reduction in mitochondrial aspect ratio within POMC neurons of female mice, with a similar trend observed in males (Fig. 6l, m, Supplementary Fig. 7l, m). These findings align with prior in vivo results[40,55] and with our earlier observations in N43/5 hypothalamic neurons, which displayed mitochondrial fragmentation after palmitate exposure in vitro (see above). In addition, mitochondrial area in POMC neurons was affected by HFD feeding only in male mice (Fig. 6n, Supplementary Fig. 7n). Interestingly, these diet-dependent alterations in mitochondrial structure within POMC neurons were attenuated in male *CerS6*[ΔPOMC] mice (Fig. 6l–n) but not to the same extent in females (Supplementary Fig. 7l–n), which is in line with

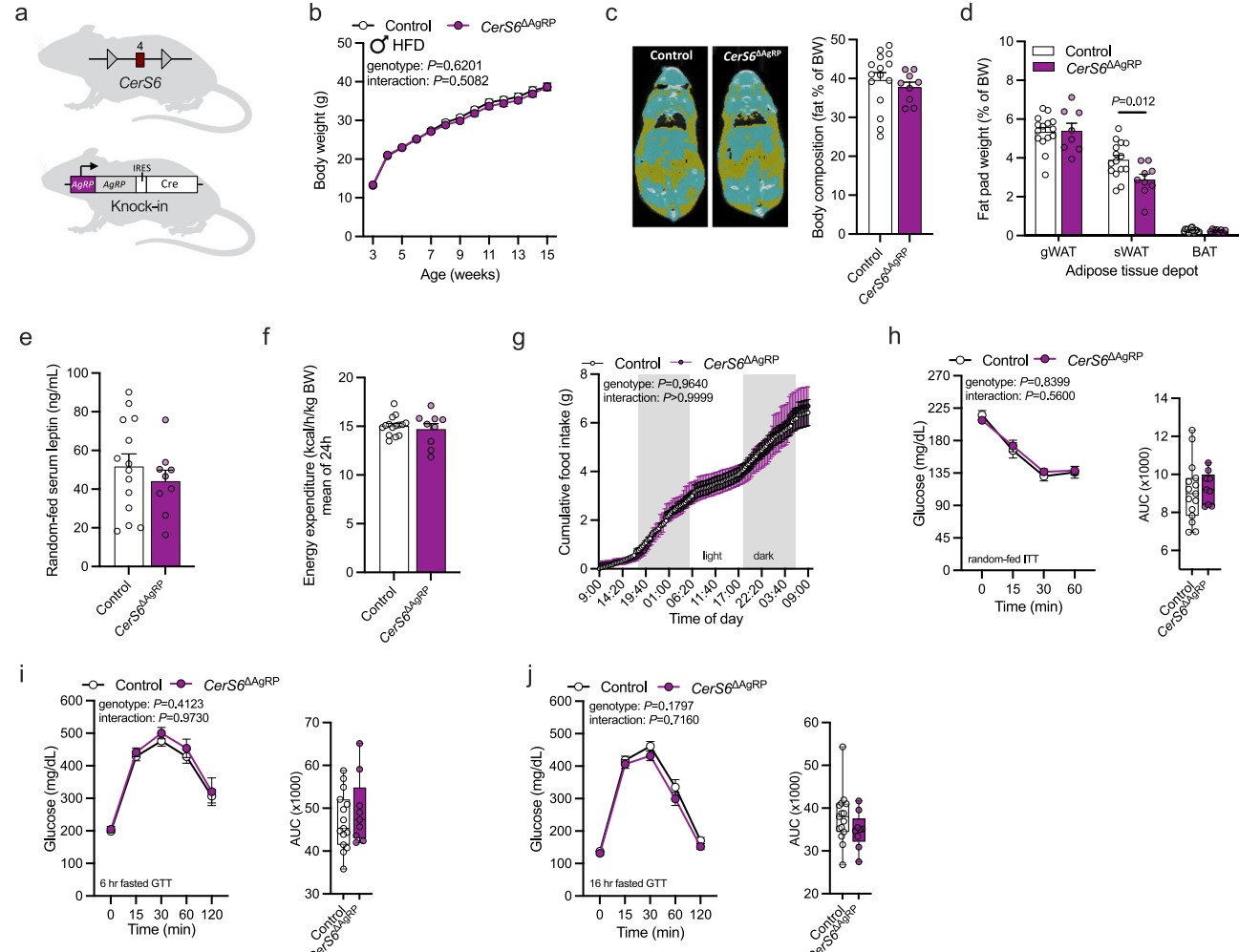

**Fig. 5 | Depletion of CerS6 in AgRP neurons does not affect glucose and energy metabolism in HFD-fed male mice. a** *CerS6*^fl/fl mice (top) in which exon 4 of *CerS6* (red square) is flanked by loxP sites (gray triangles) were bred to mice expressing Cre from the endogenous *AgRP* locus (right). The Internal Ribosome Entry Side (IRES) is used to ensure bicistronic expression of AgRP and Cre. AgRP-Cre-positive *CerS6*^fl/fl mice (*CerS6*^ΔAgRP, purple) and Cre-negative *CerS6*^fl/fl control littermates (Control, white) were used for analysis. **b** Body weight development of mice during HFD feeding (*n* = 15 vs. 9 mice/week). **c** CT scans (yellow, fat tissue; blue, non-adipose soft tissue) and body fat content relative to body weight (*n* = 15 vs. 9 mice). **d** Fat-pad weights of gonadal white adipose tissue (gWAT; *n* = 15 vs. 8 mice), sub-cutaneous WAT (sWAT; *n* = 15 vs. 9 mice), and brown adipose tissue (BAT; *n* = 15 vs. 8 mice) relative to body weight. **e** Serum leptin levels of random-fed mice (*n* = 14 vs.

9 mice). **f** Average energy expenditure over 24 h normalized for body weight (*n* = 15 vs. 9 mice). **g** Cumulative food intake over 48 h (*n* = 15 vs. 9 mice; gray background indicates dark phase). **h** Insulin tolerance test and area under the curve (AUC) for each mouse (*n* = 15 vs. 9 mice). **i, j** Glucose tolerance test following a 6 h (**i**) or 16 h (**j**) fasting period and AUCs (*n* = 15 vs. 9 mice). All data obtained from male mice. Data in (**b**–**g**) and longitudinal data in (**h**–**j**) are represented as mean values ± SEM. Boxplots indicate median ±min/max and include data points of individual mice entering the analysis. *P*-values calculated using two-tailed unpaired Student's *t*-test (**d**) or two-way RM ANOVA (longitudinal analysis in **b**, **g**–**j**). Source data and further details of statistical analyses are provided as a Source Data file. Illustration in (**a**) created with BioRender.com.

the sex-specific improvements in glucose metabolism due to POMC neuron-specific deletion of *CerS6* (see above). Notably, the dynamic regulation of mitochondrial shape in POMC neurons has been previously linked to the ability of POMC neurons to sense leptin, while obesity is associated with a shift toward mitochondrial fragmentation within POMC neurons and a reversible decrease incellular leptin sensitivity[40,56,57]. Using immunohistochemistry, we tested for phosphorylation of STAT3 at Tyr705 as a downstream readout for leptin signaling in POMC neurons of HFD-fed male mice following intraperitoneal injection of leptin at supraphysiological concentrations. While the immunoreactivity of phosphorylated STAT3 (pSTAT3) throughout the ARC was similar between the two genotypes, it was increased in POMC-positive cells in the ARC of HFD-fed *CerS6*^ΔPOMC mice compared to control animals exposed to the same diet (Fig. 6o). This observation indicates an augmented sensitivity of ARC POMC neurons to leptin in HFD-fed mice upon inhibition of CerS6-dependent ceramide synthesis.

In sum, deficiency for CerS6 in POMC neurons attenuates the diet-dependent effects on mitochondrial morphology in male mice, increases POMC neuron leptin sensitivity, and improves the insulin-dependent regulation of glucose metabolism in obesity.

## Discussion

Previous studies support the crucial role of fatty acids and lipid metabolism in the hypothalamus, influencing neuronal processes and systemic energy balance[11]. Specifically, lipid-induced inflammatory signaling in the mediobasal hypothalamus, ER stress, and mitochondrial dysfunction in neurons have been linked to the development of neuronal resistance to specific endocrine signals, such as insulin and leptin, associated with defects in glucose and energy homeostasis[58]. Saturated long-chain fatty acids can cross the blood-brain barrier and they accumulate in the hypothalamus during obesity progression[59]. Western diets and commonly used experimental HFDs are rich in

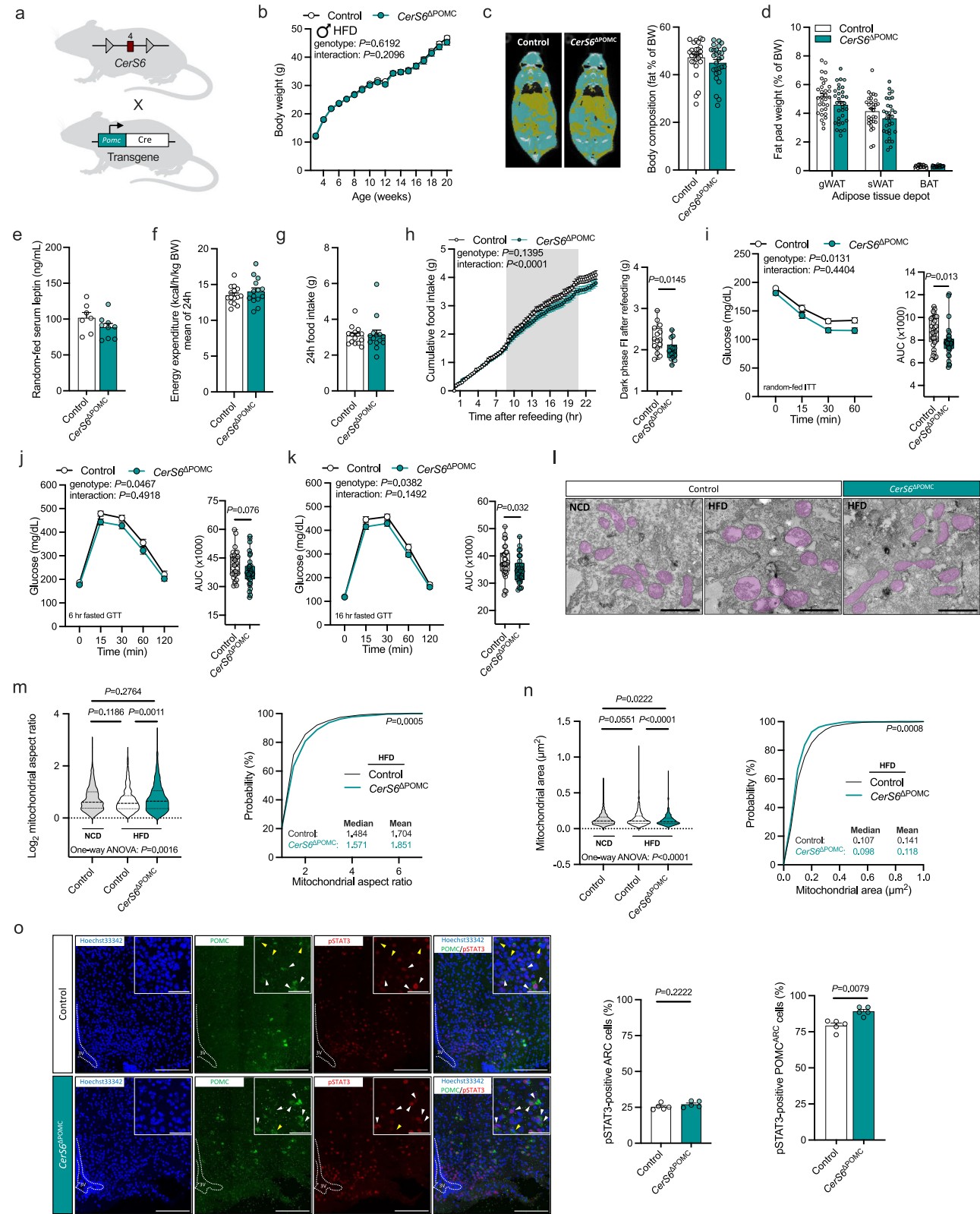

saturated fat, particularly palmitate, supplying both backbone and fatty acid chain for the synthesis of ceramides that can become cytotoxic by inducing a wide range of metabolic defects when they amass within the cell[15,21,60]. Increased hypothalamic ceramide content has been found in rodent models of overnutrition and obesity, including adult mice subjected to HFD feeding[9,13,61], early overfed rats with precocious puberty[62], and obese Zucker rats[20,21]. We report here that the

levels of several ceramide species, among them $C_{16:0}$ ceramides, increase in the hypothalamus of mouse models of obesity and diabetes, likely attributable to increased CerS6-dependent ceramide synthesis in neurons. Although $C_{16:0}$ ceramides only account for a minor proportion of the overall ceramide content in the hypothalamus, they appear to be significantly more abundant in hypothalamic neurons both in vitro and in vivo, highlighting the diversity of

**Fig. 6 | Depletion of CerS6 in POMC neurons affects mitochondrial morphology, feeding behavior, and glucose metabolism in HFD-fed male mice. a** $CerS6^{fl/fl}$ mice in which exon 4 of $CerS6$ (red square) is flanked by loxP sites (gray triangles) were bred to mice with transgenic Cre expression under control of the POMC promoter (green). POMC-Cre-positive $CerS6^{fl/fl}$ mice ($CerS6^{\Delta POMC}$, green) and Cre-negative $CerS6^{fl/fl}$ control littermates (Control, white) were used for analysis. **b** Body weight development of mice during HFD feeding ($n = 17$–$32$ vs.$16$–$33$ mice/week). **c** CT scans (yellow, fat tissue; blue, non-adipose soft tissue) and body fat content relative to body weight ($n = 27$ vs. 28 mice). **d** Fat-pad weights of gonadal white adipose tissue (gWAT, $n = 31$ vs. 32 mice), subcutaneous WAT (sWAT, $n = 31$ vs. 32 mice), and brown adipose tissue (BAT, $n = 26$ vs. 27 mice) relative to body weight. **e** Serum leptin levels of random-fed mice ($n = 7$ vs. 9 mice). **f** Average energy expenditure over 24 h normalized for body weight ($n = 15$ vs 14 mice). **g** Food intake over 24 h ($n = 15$ vs. 14 mice). **h** Cumulative food intake after refeeding following a 16-h fasting period (gray background indicates dark phase) and absolute dark phase food intake (FI) after refeeding ($n = 15$ vs.14 mice). **i** Insulin tolerance test and area under the curve (AUC) for each mouse ($n = 32$ vs. 32 mice). **j, k** Glucose tolerance test following a 6 h (**j**) or 16 h (**k**) fasting period and AUCs ($n = 32$ vs. 33 mice). **l** Transmission electron micrographs of mitochondria in ARC POMC neurons of control male mice fed a normal chow (NCD, left) or HFD (middle) for 8 weeks, and HFD-fed $CerS6^{\Delta POMC}$ mice (right; mitochondria are highlighted in purple; scale bars: 1 μm; acquired images were quantified in **m, n**). **m** Violin plots of log2-transformed mitochondrial aspect ratios (left) and cumulative distribution function (probability plot) of mitochondrial aspect ratios in HFD-fed mice (right) quantified from TEM images as represented in **l** (Control NCD: $n = 793$, Control HFD: $n = 1010$, $CerS6^{\Delta POMC}$ HFD: $n = 737$ mitochondria from 3-6 POMC neurons/mouse, 3 mice/group). **n** Violin plots (left) and probability plots (right) of mitochondrial area in HFD-fed mice quantified from TEM images as represented in **l** (Control NCD: $n = 793$, Control HFD: $n = 1010$, $CerS6^{\Delta POMC}$ HFD: $n = 737$ mitochondria from 3-6 POMC neurons/mouse, 3 mice/group). **o** Left: Confocal images of POMC (green) and pSTAT3 (red) immunoreactivity in coronal ARC brain sections of 16-h fasted HFD-fed controls (top) and $CerS6^{\Delta POMC}$ mice (bottom), 45 min after injection of leptin (6 mg/kg ip; white arrows indicate cells positive for POMC and pSTAT3, yellow arrows indicate cells positive for POMC and negative for pSTAT3; scale bars: 150 μm for overview images and 50 μm for zoom images; blue staining depicts nuclei stained by Hoechst33342). Right: quantification of pSTAT3-positive cells in the ARC relative to all imaged ARC cells (left) or pSTAT3/POMC-positive cells relative to all imaged POMC-positive cells in the ARC (right; $n = 5$ vs. 5 mice). All data obtained from male mice. Data in (**b**–**g**, **o**) and longitudinal data in (**h**–**k**) are represented as mean values ± SEM. Boxplots indicate median ±min/max and include data points of individual mice entering the analysis. Dashed lines in the violin plots indicate median and dotted lines indicate the first and third quartile, respectively. *P*-values calculated using two-tailed unpaired Student's *t*-test (dark phase FI in **h**, AUC in **i**–**k**), two-tailed Wilcoxon–Mann–Whitney test (**o**), one-way ANOVA followed by Tukey's multiple comparison test (**m, n**), Kolmogorov–Smirnov test (distributions in **m, n**), two-way RM ANOVA (longitudinal analysis in **h**, **i**–**k**), or a mixed-effects model (**b**). Source data and further details of statistical analyses are provided as a Source Data file. Illustration in (**a**) created with BioRender.com.

ceramide distribution patterns between different hypothalamic cell types. In previous studies, stimulation or inhibition of general ceramide synthesis in the central nervous system of obese animal models showed that brain ceramides play crucial roles in the deterioration of metabolic health[20,21]. The present study reveals that the adverse metabolic consequences of hypothalamic ceramide accumulation in obesity depend on endogenous ceramide production, specifically by CerS6, in distinct neuronal cell types of the hypothalamus. CerS6-derived ceramides impinge on pathways associated with ER homeostasis and mitochondrial dynamics and function, leading to alterations in neuronal leptin sensitivity and systemic metabolic control.

Fatty acids in the hypothalamus differentially affect metabolic processes depending on their respective acyl chain lengths[5]. Icv infusion of oleate ($C_{18}$) inhibits food intake and decreases hepatic glucose production, whereas infusion of palmitate ($C_{16}$) induces neuroinflammation and ARC leptin resistance, affecting feeding behavior and compromising systemic insulin sensitivity[63,64]. Similarly, our results demonstrate a ceramide acyl chain-specific regulation of glucose and energy metabolism through effects in hypothalamic neurons. In HFD-fed mice, hypothalamic neuron-specific inhibition of $C_{16:0}$ ceramide synthesis by inactivation of CerS6, but not of $C_{18:0}$ ceramide synthesis by inactivation of CerS1, increases energy expenditure, reduces adiposity, and improves the insulin-dependent regulation of glucose metabolism. Thus, our experiments provide evidence for a critical role, specifically of $C_{16:0}$ ceramides, within hypothalamic neurons in the pathogenesis of metabolic deterioration. This aligns with the findings in a previous study, in which central administration of the $C_6$ ceramide analog stimulated hypothalamic $C_{16:0}$ but not $C_{18:0}$ ceramide formation in rats, sufficient to provoke inflammatory marker expression, ER stress in the mediobasal hypothalamus, and a feeding-independent increase in adiposity[20]. Still, despite the lack of a discernable metabolic phenotype in HFD-fed $CerS1^{\Delta Nkx2.1}$ mice, the data presented here do not exclude the possibility that hypothalamic $C_{18:0}$ ceramides originating from an alternative source, e.g., CerS4-dependent ceramide synthesis, could exert additional functions.

In addition to $CerS6$, hypothalamic mRNA expression of the alternative $C_{16:0}$ ceramide-producing $CerS5$ was increased in obese db/db and HFD-fed mice. This suggests potential metabolic effects of CerS5-derived $C_{16:0}$ ceramides in the hypothalamus. However, our in vitro studies showed that palmitate excess does not affect $CerS5$

expression in N43/5 cells, implying that hypothalamic $CerS5$ in vivo is modulated by other obesity-related factors independent of palmitate. Single-cell sequencing results further indicate $CerS5$ expression in various hypothalamic cell types, while $CerS6$ expression is mainly found in neurons, suggesting that the obesity-related increase in hypothalamic $CerS5$ mRNA could be due to regulation in non-neuronal cells rather than neurons, as opposed to $CerS6$. In addition, as previously shown, the conventional deletion of $CerS6$, but not $CerS5$, confers protection against diet-induced obesity and preserves glucose homeostasis in mice, attributing a less critical role to CerS5- compared to CerS6-dependently formed $C_{16:0}$ ceramides in the etiology of metabolic disorders[27]. It should be taken into account that CerS6 may also use $C_{14}$ fatty acyl CoA as a substrate for ceramide formation. Although $C_{14:0}$ ceramides were present only at low levels in hypothalamic neurons and were not reduced in the hypothalamus of conventional CerS6-deficient mice, their potential role in regulating systemic energy balance cannot be entirely excluded.

By targeting CerS6 in subpopulations of hypothalamic neurons, we identified a role of CerS6-dependent ceramide synthesis in SF-1 neurons of the VMH and POMC neurons of the ARC with respect to metabolic control, but not in AgRP neurons. Previous studies have already proposed a specific involvement of de novo synthesized ceramides in cells of the VMH and ARC. However, there has been no direct investigation into the de novo synthesis pathway of ceramides in specific neurons to validate this hypothesis. In POMC neurons, the experimental manipulation of sphingolipid metabolism was confined to the formation of more complex sphingolipids, where inhibition of ceramide glycosylation via POMC neuron-specific deletion of glucosylceramide synthase (GCS) led to weight gain and obesity, even under normal diet conditions[65]. We show here that blocking a specific branch of ceramide synthesis through deficiency for CerS6 in SF-1- and POMC-expressing cells is sufficient to improve systemic insulin sensitivity in male mice, similar to what we observed in mice with ablation of CerS6-dependent ceramide formation in a larger proportion of hypothalamic neurons.

In this study, we observed sex-specific metabolic phenotypes when inhibiting CerS6-dependent ceramide synthesis in different hypothalamic neurons. Specifically, HFD-fed female mice with CerS6 deficiency in SF-1 neurons showed a feeding-independent reduction in adiposity but only moderate alterations in glucose homeostasis as

compared to control mice exposed to the same diet. In contrast, HFD-fed $CerS6^{\Delta SF-1}$ male mice showed reduced food intake and clearer improvements in insulin sensitivity and glucose tolerance, despite unaltered adiposity. These findings align with previous studies indicating that estradiol regulates energy balance by modulating hypothalamic ceramide levels in female rats[66]. The estrogen receptor (ER)α is expressed in specific neurons of the VMH where they determine a sexually dimorphic regulatory node of energy expenditure[67]. Estradiol ameliorates ceramide-induced lipotoxicity and ER stress in the VMH, associated with a reduction in body weight in female rats independent of food intake[66]. Our study implies that CerS6-dependent ceramide synthesis might be a critical determinant in this process, although it's worth noting that we did not observe significant effects of CerS6 deletion in SF-1-expressing cells of female mice concerning basal or cold-induced energy expenditure.

In future studies, it will be interesting to approach whether CerS6-dependent ceramide synthesis affects the homeostatic control of energy metabolism and additional processes also in other areas, neurons, and cell types of the hypothalamus than those investigated here. This is particularly relevant as none of the male mouse models of cell-type-specific CerS6 deficiency recapitulate the reduction in adiposity observed in male mice with nearly neuron-wide ablation of CerS6 in the hypothalamus. This suggests additive effects of CerS6 deficiency in SF-1 and POMC neurons and/or critical metabolic functions of CerS6 in other hypothalamic neurons that were not subject to this study. Along these lines, a significant role of ceramides in the paraventricular nucleus of the hypothalamus (PVH) in obesity-associated precocious puberty has been identified recently[62].

In the N43/5 in vitro model of hypothalamic neurons, we observed that CerS6-dependent ceramide synthesis regulates palmitate-induced stress responses of the ER and mitochondria. ER stress is triggered by the accumulation of unfolded proteins within the ER lumen, stimulating an integrated unfolded protein stress response (UPR) to limit the disturbances in internal homeostasis[68]. However, in obesity, this adaptive response is insufficient to restore ER functionality, resulting in loss of cellular integrity and the development of metabolic disease[69]. The UPR involves expression of ATF4 and splicing of $Xbp1$, closely linked to metabolic regulation in the hypothalamus[36,70]. Here, we demonstrate that CerS6-dependent $C_{16:0}$ ceramide synthesis in cultured hypothalamic neurons promotes palmitate-induced $Xbp1$ splicing as well as expression of ATF4 and the ER chaperone GRP78/BiP. These findings suggest that elevated hypothalamic ER stress in obese mice may arise from increased CerS6-dependent ceramide synthesis in hypothalamic neurons due to fatty acid excess. Prior research on site-directed SPTLC1-2 overexpression, which promotes ceramide synthesis, also supports the contribution of aberrant ceramide production in the VMH to ER stress[71]. Moreover, our data align with studies showing that VMH-specific overexpression of the ER chaperone GRP78/BiP can alleviate metabolic deficits induced by $C_{16:0}$ ceramide synthesis via icv injection of ceramide analogs[20]. Our observation that inhibiting CerS6-dependent ceramide synthesis in SF-1 and POMC neurons improves glucose homeostasis during HFD feeding, particularly in male mice, underscores the critical role of CerS6-derived ceramides in obesity-associated ER stress in these types of neurons.

ER stress reciprocally affects mitochondria, associated with mitochondrial stress and dysfunction[72]. Similar to the ER, mitochondria exert control over an integrated UPR machinery ($UPR^{mt}$) to counter internal proteotoxic stress, essential for maintaining insulin sensitivity of the ARC[73,74]. We found here that CerS6-dependent $C_{16:0}$ ceramide synthesis promotes palmitate-driven mitochondrial stress and mitochondrial fragmentation in cultured hypothalamic neurons and upon HFD-feeding in POMC neurons in vivo. Indeed, mitochondrial stress is closely linked to changes in morphology, regulated by opposing events of membrane fusion and fission to ensure dynamic adaption of mitochondrial and cellular homeostasis to metabolic

inputs and nutritional switches[75]. In diet-induced obesity, POMC neurons exhibit ER stress, loss of mitochondria/ER contact sites, altered mitochondrial cristae morphology, and a switch toward mitochondrial fission[40,55]. Failure of POMC neurons to adapt mitochondrial dynamics and bioenergetics to metabolic cues has been linked to the development of leptin resistance and impaired metabolic control[40,56,76]. Similarly, SF-1-expressing cells in the VMH depend on regulated mitochondrial dynamics for neuronal and systemic glucose homeostasis[77]. Our findings suggest that CerS6-derived ceramides in ARC POMC and VMH SF-1 neurons may modulate these metabolic processes by inducing ER and mitochondrial stress and altered mitochondrial dynamics. Accordingly, the protection from HFD-induced alterations in the mitochondrial network of POMC neurons in $CerS6^{\Delta POMC}$ mice, as we observed in male but not female animals, was associated with increased responsiveness of POMC neurons for leptin-induced phosphorylation of STAT3 at the Tyr705 residue.

Of note, N43/5 cells used in this study have an XX-karyotype, and this in vitro model largely resembles what is detected in POMC neurons of male $CerS6^{\Delta POMC}$ mice, arguing that the sex-differences in ER/mitochondrial stress regulation in vivo rather originate from differences in sex hormones than from sex-specific cell-intrinsic regulation. Importantly, ER/mitochondrial crosstalk and mitochondrial dynamics are crucial for intracellular $Ca^{2+}$ signaling and neuronal excitability, while mitochondrial $Ca^{2+}$ handling of POMC neurons is altered in diet-induced obese mice that dramatically decreases POMC neuron activity[53]. Previous experiments had also indicated a specific role of CerS6 in mitochondrial $Ca^{2+}$ handling in oligodendrocytes[78]. Thus, it is conceivable that CerS6-derived ceramide-dependent effects on ER/mitochondrial dynamics during the development of diet-induced obesity lead to impaired ER/mitochondrial $Ca^{2+}$ homeostasis, which alters POMC neuron intrinsic activity, ultimately changing feeding behavior and impairing metabolic control.

Concerning the ceramide-dependent regulation of mitochondrial dynamics in hypothalamic neurons, a mechanism similar to our earlier findings in the liver of mice might be at play[27]. We previously found that CerS6-dependent sphingolipids directly target the mitochondrial fission machinery to modulate mitochondrial morphology, mitochondrial respiratory activity, and glucose homeostasis in obesity[27]. Another study recently reported on a potential relation between hypothalamic ceramides and mitochondrial dynamics in metabolic control[79]. Specifically, the authors discovered that the synthetic sphingolipid SH-BC-893 could inhibit palmitate- and ceramide-induced mitochondrial fragmentation in vitro, preserving mitochondrial function and preventing ER stress[79]. In obese mice, oral administration of SH-BC-893 acutely improved leptin sensitivity by reversing ceramide-induced mitochondrial fragmentation in ARC cells, reducing food intake, and correcting diet-induced obesity and its metabolic consequences, similar to what we found upon deletion of $CerS6$ in POMC neurons[79]. Our findings indicate that CerS6-dependent ceramide synthesis promotes ER stress and alters mitochondrial integrity in hypothalamic neurons in response to prolonged exposure to fatty acids, contributing to systemic metabolic damage in obesity.

An important question remaining relates to the physiological roles of ceramides in the hypothalamus regarding basal metabolic rate. Ceramides may act as nutrient sensors in a cell-autonomous fashion, accumulating under fatty acid overload to balance energy intake, storage, and expenditure[14]. Ceramides in the hypothalamus are proposed to mediate the orexigenic effects of ghrelin and oppose the anorectic effects of leptin, regulating appetite and food intake as second messenger molecules[80,81]. We found that disruption of CerS6-dependent ceramide synthesis in specific hypothalamic neurons influences feeding behavior and energy expenditure in obese mice. This supports the idea that CerS6-derived $C_{16:0}$ ceramides in the hypothalamus control the dynamic regulation of energy balance, responding to nutrient excess or deficit. Along these lines, CerS6-

 

dependent ceramide synthesis controls ER/mitochondrial stress and mitochondrial dynamics, potentially to initiate an integrated stress response in order to protect neurons from lipotoxic damage and adjust mitochondrial morphology in the regulation of neuronal plasticity responding to fatty acid fluctuations. This also aligns with a current theory according to which the adverse metabolic effects of ceramide accumulation in different cell types result from an initial adaptive response to increased fatty acid flux that fails under the sustained burden of HFD feeding, rather than from ceramide function per se. Further investigations are warranted to explore these hypotheses.

The study highlights the crucial role of the acyl chain length ceramide-specific regulation of metabolic homeostasis in hypothalamic neurons. CerS6 emerges as a key player in this process, suggesting its involvement in appetite and metabolic rate control during nutritional switches. Overeating with a HFD elevates CerS6 and $C_{16:0}$ ceramide content in the hypothalamus, leading to impaired systemic glucose metabolism and energy balance. Targeting CerS6-dependent ceramide synthesis in the hypothalamus holds promise for obesity and diabetes therapies, offering potential advantages over complete inhibition of sphingolipid/ceramide synthesis with reduced side effects.

## Methods
### Animal care
All animal procedures were conducted in compliance with protocols approved by local government authorities (Bezirksregierung Köln). Permission to maintain and breed mice as well as for all experimental protocols in this study was issued by the Department for Environment and Consumer Protection · Veterinary Section, Köln, North Rhine-Westphalia, Germany (84-02.04.2014.A037, 84-02.04.2016.A520, and 81-02.04.2019.A252). Mice at a maximum count of five per cage were housed in individually ventilated cages (IVCs) at 22 °C–24 °C ambient temperature using a 12 h light/12 h dark cycle, with humidity of 50–70%. Animals had access to water and food *ad libitum*. Food was only withdrawn if required for an experiment during defined fasting periods. From 3 or 4 weeks of age onward, mice were fed either a normal chow diet (NCD; R/M-H; Ssniff Diet) containing 39% carbohydrates, 19.3% protein, 3.3% fat (9% calories from fat), a high-fat diet (HFD; EF D12492-(I); Ssniff Diet) containing 25.3% carbohydrates, 24.1% protein, and 34.6% fat (60% of calories from fat), or a control to HFD diet (CD; EF D12450B; Ssniff Diet) containing 58.1% carbohydrates, 18.1% protein, and 5.1% fat (13% of calories from fat) as mentioned in the manuscript where appropriate.

### Experimental models
**Mouse lines.** C57BL/6 N male mice (strain code: 027) were obtained from Charles River Laboratories at 4 weeks of age and fed either a CD or HFD (see above). C57BL/6 N male mice, employed for the assessment of hypothalamic CerS expression and sphingolipid content (Fig. 1b-f, Supplementary Fig. 1a-c, e-g), unterwent intravenous administration of an adeno-associated virus serotype 8 (AAV8) carrying a liver-specific construct during the course of another study. We exclusively utilized those mice for experiments that had received the control AAV8, i.e., AAV8-TBG-eGFP (here: used for qPCR analysis; Fig. 1b) or AAV8-scrambled shRNA (here: used for Western blot and lipidomic analysis; Fig. 1c-f, Supplementary Fig. 1e, f), which we considered to not interfere with hypothalamic CerS expression and ceramide levels. These animals received the CD or HFD for 16 weeks, respectively. Male db/db mice and misty controls were purchased from The Jackson Laboratory. All other animal models employed in this study were derived from in-house breedings. Generation and verification of CerS6[Δ/Δ], CerS1[fl/fl], and CerS6[fl/fl] mice has been detailed in previous publications[25,29]. CerS1[fl/fl] and CerS6[fl/fl] mice were crossed with those expressing the Nkx2.1-Cre transgene[43] for ablation of CerS1 or

CerS6 in a large proportion of hypothalamic neurons. In addition, CerS6[fl/fl] animals were bred to previously described SF-1-Cre-[82], AgRP-IRES-Cre-[83], and POMC-Cre-expressing mice[51] for cell-type-specific targeting of CerS6. To obtain experimental cohorts, breeding pairs were set up, with both mice in a pair carrying homozygous floxed CerS alleles. Additionally, one of the mice in each pair expressed the Cre recombinase under the control of the corresponding cell-type-specific promoter. CerS[fl/fl] Cre-positive mice were employed as the test group and CerS[fl/fl] Cre-negative littermates served as the control group. After weaning, individual mice were marked by ear punching, and tissue obtained was utilized for DNA isolation and subsequent genomic PCR analysis (performed in-house or by GVG Genetic Monitoring, Leipzig) using the following primers:

For genotyping of CerS1-flox alleles:
1: TATCCAGTGGCGTCTTTGTG
2: TCAGGTTCTGGAAGGGAATG
For genotyping of CerS6-flox alleles:
1: ATTCAGTATGGTGCCAGCAAAGC
2: GAGACTGGAAAGTAGTGACCAATCC
For genotyping of the Nkx2.1-Cre and POMC-Cre transgenes
1: CTGCAGTTCGATCACTGGAAC
2: AAAGGCCTCTACAGTCTATAG
3: TCCAATTTACTGACCGTACA
4: TCCTGGCAGCGATCGCTATT
For genotyping of the AgRP-IRES-Cre knockin:
1: GGGCCCTAAGTTGAGTTTTCCT
2: GATTACCCAACCTGGGCAGAAC
3: GGGTCGCTACAGACGTTGTTTG
For genotyping of the SF-1-Cre transgene
1: GGTCAGCCTAATTAGCTCTGT
2: GATCTCCAGCTCCTCCTCTGTC
3: TGCGAACCTCATCACCACTCGTTGCAT
4: CTGAGCTGCAGCGCAGGGACAT
For genotyping of CerS1 exon 2 deletion:
1: AAACTCATGGCAATCCTCCTGCC
2: TATCATGGTCCCTAGCAAGGAGCC
For genotyping of CerS6 exon 4 deletion:
1: ATTCAGTATGGTGCCAGCAAAGC
2: AGCAAACTCTGCTGGCAAGATTA
3: GGATTGGTCACTACTTTCCAGTCTC

**Cell lines.** For in vitro experiments, we employed the immortalized embryonic mouse hypothalamus cell line mHypoE-N43/5, which has been described previously[34].

### Cell Culture, RNA interference, and palmitate treatment
N43/5 cells were maintained in Dulbecco's modified Eagle's medium (DMEM) (Life Technologies) containing 4.5 g l[-1] glucose, supplemented with 10% fetal calf serum (FCS, PAN-Biotech), 1 mM sodium pyruvate (Gibco), 2 mM L-Glutamine (Gibco), 100 μM non-essential amino acids (Gibco), and 1% penicillin/streptomycin (Gibco) at 37 °C with 5% $CO_2$. Cell numbers were routinely monitored using an EVE automatic cell counter (NanoEntek).

For RNA interference (RNAi) experiments, cells were seeded into the appropriate cell culture dish: a 6-well-plate (for protein and RNA isolation), a 10 cm dish (for lipidomic analysis), or an XF96 cell culture microplate (Agilent Technologies; for assessment of mitochondrial respiration). Cells were seeded 24 h before treatment to achieve ~60–70% confluency on the day of the assay. Opti-MEM reduced serum medium (Gibco; Thermo Scientific, Cat# 31985062) and Lipofectamine RNAiMAX transfection reagent (Invitrogen; Thermo Scientific, Cat# 13778150) were used for the transfection of *Silencer* select small interfering RNAs (siRNA, Ambion). Transfection was conducted for a duration of 48 h at 37 °C in accordance with the manufacturer's protocol. The following siRNAs have been used:

- CerS6 siRNA: ID s109527; sequence 5′–GGAUGUUCGGAGCA UUCAAtt–3′; Cat# 4390816
- Negative control #1 siRNA: Cat# 4390843

For palmitate treatment, 500 μM of palmitic acid (Sigma-Aldrich, Cat# P0500) were conjugated to 1% (w/v) fatty acid-free low-endotoxin BSA (Sigma-Aldrich, Cat# A8806) in DMEM containing all supplements at 37 °C for 20–30 min. Subsequently, cells were cultured in the presence of BSA-conjugated palmitate (PAL) or the BSA vehicle, either for the indicated duration or, if not specified, for a period of 9 h at 37 °C in a 5% $CO_2$ environment. Afterward, cells underwent different preparatory steps: cells were either used for mitochondrial respiration analysis, fixed within the culture dish and prepared for TEM, or frozen in liquid nitrogen within the dish and stored at -80 °C for subsequent RNA or protein extraction. Alternatively, cells were gently scraped into ice-cold PBS, centrifuged at 5000xg for 10 min at 4 °C, shock-frozen in liquid nitrogen, and preserved at -80 °C for future lipid extraction (all procedures described below).

## Assessment of mitochondrial respiration in N43/5 neurons

N43/5 cells were seeded into Seahorse Bioscience XF96 cell culture microplates (Agilent, Cat# 102416-100) at a starting density of 8000 cells per well. On the next day, cells were treated with the respective siRNA for 48 h, followed by exposure to palmitate for 9 h, as detailed above. One day prior to the assay, cartridges were hydrated with calibrant solution (Agilent) at room temperature in the dark. On the day of the experiment, even distribution of the cells and consistent adherence to the well was verified using a microscope (Leica DM IL LED). Cell culture media was changed to XF Base Medium (Agilent) supplemented with 1 mM glutamine and calibrated at pH 7.4. Oxygen consumption rates (OCR) were measured using the XFe96 Extracellular Flux Analyzer operated by Seahorse Wave Desktop software v2.4.1 (Agilent) according to the manufacturer's instructions. OCR was recorded in four baseline measurement cycles and following the injection of 10 mM glucose, 1 μM oligomycin, 1 μM FCCP, and 1 μM Rotenone/Antimycin A, respectively (XF Cell Mito Stress Test Kit, Agilent, Cat#103015-100). After the Seahorse analysis, wells were normalized using the CyQUANT Cell Proliferation Assay (Invitrogen, Cat# C7026). Normalized OCR data was analyzed using the Multi-File Seahorse XF Cell Mito Stress Test Report Generator (Agilent).

## PCR of genomic DNA

PCR was performed using template DNA extracted from either ear punches (general assessment of the mouse genotype), tail biopsies, or hypothalamus tissue samples (assessment of tissue-specific Cre-mediated recombination). For resection of the entire hypothalamus, brains were extracted from euthanized mice. Subsequently, 2 mm coronal brain sections that encompass the hypothalamus were sliced using a stainless-steel mouse brain matrix. Dissection of the hypothalamus was carried out under a microscope (SteREO Discovery.V12, Zeiss) using sharp razor blades. ARC- and VMH-containing tissue samples were taken from 2 mm thick coronal brain sections according to the Allen Brain Atlas using 1 mm stainless steel biopsy punches (Rainer Medizintechnik). After excision of the ARC and VMH, the hypothalamic tissue situated superior to the VMH was collected and designated as the "rest of hypothalamus". DNA from tissue samples was extracted at 55 °C overnight in 100 mM Tis-HCl pH 8.5, 5 mM EDTA pH 8.0, 0.2% SDS, and 200 mM NaCl supplemented with proteinase K (1:100). DNA was precipitated with isopropanol. DNA pellets were washed once in 70% EtOH, air dried at room temperature, and resuspended in nuclease-free water. To evaluate the deletion of exon 4 in *CerS6* or exon 2 in *CerS1*, the designated primers were used as indicated above. All amplifications were performed in a total reaction volume of 25 μL containing template DNA, 25 pmol of each primer, 25 μM dNTP mix, 1x DreamTaq Green Buffer, and 1 unit DreamTaq DNA polymerase

(Thermo Scientific). PCRs were run on a PCR cycler (FlexCycler², Analytic Jena). PCR-amplified DNA fragments were applied to 2% (w/v) agarose gels with MidoriGreen (1:20000) and electrophoresed at 150 V. Gels were imaged with a FastGene FAS-V Imaging System (Nippon Genetics Europe).

## Insulin tolerance test (ITT)

ITTs were performed on random-fed mice at 11 weeks of age ($CerS6^{\Delta SF-1}$, $CerS6^{\Delta AgRP}$, $CerS6^{\Delta POMC}$ mice), 13 weeks of age ($CerS1^{\Delta Nkx2.1}$, $CerS6^{\Delta Nkx2.1}$ mice), and at 16 weeks of age (CD/HFD-fed C57BL/6 N mice). Prior to the experiment, animals were placed in fresh cages and had no access to food during the experiment. Blood glucose concentrations were measured from whole venous blood using an automatic glucose monitor (Contour XT, Bayer Healthcare). Following the determination of body weight and basal blood glucose levels, mice received an intraperitoneal (IP) injection of 0.75 U/kg body weight human insulin (Lilly Pharma) dissolved in 0.9% saline, and blood glucose concentrations were monitored at 15, 30, and 60 min after insulin injection.

## Glucose tolerance test (GTT)

GTTs were performed on mice fasted for 6 h during the light-phase (6 h fasted GTT) or 16 h overnight (16 h fasted GTT) at 12–15 weeks of age ($CerS1^{\Delta Nkx2.1}$, $CerS6^{\Delta Nkx2.1}$, $CerS6^{\Delta SF-1}$, $CerS6^{\Delta AgRP}$, $CerS6^{\Delta POMC}$ mice) or 17 weeks of age (CD- and HFD-fed C57BL6/N mice). Following the determination of basal blood glucose levels, mice were injected IP with 20% glucose (10 mL/kg body weight, bela-pharm) and blood glucose was monitored at 15, 30, 60, and 120 min after glucose injection. Food was withdrawn throughout the experimental period. For $CerS6^{\Delta SF-1}$, $CerS6^{\Delta AgRP}$, and $CerS6^{\Delta POMC}$ mice, glucose levels at each timepoint represent the mean of at least two repetitive measurements using two independent glucose monitors.

## Indirect calorimetry and food intake

Metabolic measurements were obtained using the PhenoMaster (TSE systems), an open circuit calorimetry system. Mice at 16–21 weeks of age were acclimatized to the metabolic chambers prior to the analysis. The ambient temperature was consistently maintained at 22 °C and was reduced to 6 °C for the purpose of analyzing energy expenditure during cold exposure in $CerS6^{\Delta SF-1}$ mice. Food and water were provided *ad libitum* in the appropriate devices, and food intake was measured by the integrated automatic instruments. For the analysis of food intake following fasting and refeeding, the bedding was changed, and food hoppers were removed overnight for 16 h. Subsequently, food hoppers were reinstated in the cage the following day, and food intake was monitored for the ensuing 24 h. Measurements were taken every 20 min. Data points exhibiting excessive fluctuations in food intake, defined as increments of 2 g or more per 20 min, where manually excluded from the analysis.

## Analysis of body composition

Fat mass of $CerS6^{\Delta Nkx2.1}$, $CerS1^{\Delta Nkx2.1}$, and respective control mice was determined via nuclear magnetic resonance (NMR; Analyzer minispeq mq7.5, Bruker Optik) in awake mice at 20 weeks of age. Fat mass of $CerS6^{\Delta SF-1}$ (at 19 weeks of age), $CerS6^{\Delta AgRP}$ (at 17 weeks of age), $CerS6^{\Delta POMC}$ (at 20 weeks of age), and respective control mice was analyzed by micro-computed tomography (micro-CT)-based imaging of isoflurane-anesthetized mice. In that case, data acquisition was performed in an IVIS Spectrum CT-scanner (Caliper LifeSciences) using the IVIS LivingImage Software V4.3.1. Quantification of fat mass was performed using a modified version of the Vinci software package 4.61.0.

## Protein extraction and Western blot analysis

Hypothalamus samples were lysed with ceramic beads in tissue lysis buffer (50 mM Tris, 130 mM NaCl, 5 mM EDTA, 1% NP-40, 1 mM PMSF, 1 mM NaF) supplemented with 1x Halt protease and phosphatase

inhibitor cocktail (Thermo Scientific, Cat# 78446), using the vibrating tube mill MM 400 (Retsch) equipped with the TissueLyser Adapter Set (Qiagen). Frozen cell pellets were lysed in 1x cell lysis buffer (Cell Signaling, Cat# 9803) supplemented with 1 mM PMSF and 1x Halt protease and phosphatase inhibitor cocktail (Thermo Scientific, Cat# 78446). Lysis was carried out by pipetting the lysate up and down using a 0.2 mL pipette, and then subjecting it to vortexing, rotation for 1 h at 4 °C, and brief sonication. Protein quantification of lysates was performed by a colorimetric assay using BCA Protein Assay Kit (Pierce). Absorption spectroscopy at 595 nm was conducted using a FilterMax F5 Multi-Mode microplate reader and SoftMax Pro 6.3 software (Molecular Devices). Protein concentrations were calculated with a BSA standard curve. Equal amounts of total protein per sample were subjected to protein separation via SDS-PAGE using 4-15% TGX Precast Midi Protein Gels (26 well, Criterion) and transferred to PVDF membranes using a Perfect-Blue 'Semi-Dry' Blotter (Peqlab). Membranes were blocked with 5% (w/v) nonfat dried milk (AppliChem, Cat# A0830,0500) dissolved in Tris-buffered saline (TBS) for at least 30 min, followed by the incubation with primary antibodies dissolved in 2.5% (w/v) milk in TBS containing 0.1% Tween-20 (TBST) overnight at 4 °C. The following day, secondary antibodies conjugated to horseradish peroxidase (HRP) dissolved in 2.5% (w/v) milk in TBST were incubated for 1–2 h at room temperature. After each incubation step with primary or secondary antibody, membranes were washed 3x in TBST. Membranes were briefly incubated with SuperSignal ECL Western Blotting Substrate (Thermo Scientific, Cat# 34076), and luminescence was detected using a Fusion Solo imaging system and FusionCapt Advance software (Vilber). If required, membranes were stripped in Stripping Solution (62.5 mM Tris pH 6.8, 2% (w/v) SDS, 0.7% (v/v) β-mercaptoethanol) for 20–30 min at 56 °C before re-probing. Densitometric analyses of immunoblots were performed with the Fiji software package.

The following antibodies were used for immunoblot analysis: Mouse monoclonal anti-CerS6 (5H7) (Abnova Cat# H00253782-M01, RRID:AB_489924, dilution 1:1000), Rabbit monoclonal anti-ATF4 (D4B8) (Cell Signaling Technology Cat# 11815, RRID:AB_2616025, dilution 1:1000), Rabbit polyclonal anti-Calnexin (Millipore Cat# 208880, RRID:AB_2069031, dilution 1:5000), Mouse monoclonal anti-ACTIN (AC15) (Sigma-Aldrich Cat# A5441, RRID:AB_476744, dilution 1:10000), Goat polyclonal GRP78/BiP (N-20) (Santa Cruz Biotechnology Cat# sc1050, RRID:AB_631616, dilution 1:1000), Goat polyclonal anti-rabbit IgG-peroxidase (Sigma-Aldrich Cat# A0545, RRID:AB_257896, dilution 1:2000), Goat polyclonal anti-mouse IgG-peroxidase (Sigma-Aldrich Cat# A4416, RRID:AB_258167, dilution 1:2000), Mouse monoclonal anti-goat/sheep IgG-peroxidase (GT34) (Sigma-Aldrich Cat# A9452, RRID:AB_258449, dilution 1:2000).

## RNA extraction, reverse transcription, and quantitative PCR

Total RNA was extracted using Qiazol reagent according to the RNeasy Lipid Tissue Kit protocol (Qiagen, Cat# 74804), including on-column DNAse digestion of all samples with the RNase-free DNase set (Qiagen, Cat# 79254). Ceramic beads were used for tissue disruption, and samples were homogenized in 1 mL Qiazol reagent using the vibrating tube mill MM 400 (Retsch) equipped with the TissueLyser Adapter Set (Qiagen). Cells were frozen at −80 °C, scraped from the dish in 1 mL Qiazol, and homogenized with 10 strokes using a 1000 μL pipette. Chloroform was added for phase separation, and total RNA was precipitated from the aqueous supernatant using 70% EtOH. RNA was captured on RNeasy Mini spin columns (Qiagen) and eluted using nuclease-free water. RNA concentrations were determined using a NanoDrop ND-1000 UV/Vis-spectrophotometer and software ND-1000 v3.8.1 (PeqLab). RNA (2000 ng) was reverse-transcribed with High-Capacity cDNA RT Kit and amplified with TaqMan Universal PCR-Master Mix and Assay-on-demand kits (Applied Biosystems), or with SYBR Green Master Mix (Applied Biosystems) and primers synthesized

by Eurogentec. Quantitative PCR was performed on a QuantStudio 7 Flex Real-Time PCR System using the QuantStudio Real-Time PCR Software v1.7.1 (Life Technologies). Relative expression of samples was adjusted for total RNA content and normalized to the mRNA expression levels of *hypoxanthine-guanine phosphoribosyl transferase* (*Hprt*) and/or *beta-glucuronidase* (*Gusb*), or the *TATA-binding protein* (*Tbp*). Calculations were performed by a comparative method (as $2^{-\Delta\Delta CT}$). The following TaqMan qPCR probes (Applied Biosystems, purchased from Thermo Scientific) were used:

*CerS1* (Mm00433562_m1), *CerS2* (Mm00504086_m1), *CerS3* (Mm03990709_m1), *CerS4* (Mm00482658_m1), *CerS5* (Mm01305570_m1), *CerS6* (Mm01270928_m1), *Atf4* (Mm0515324_m1), *Clpp* (Mm00489940_m1), *Fgf21* (Mm00840165_g1), *Ddit3* (Mm00492097_m1), *Hspa5* (MM00517691_m1), *Gdf15* (Mm00442228_m1), *Tbp* (Mm01277042_m1), *Hprt* (Mm01545399_m1), *Gusb* (Mm00446953_m1).

For the detection of both unspliced and spliced *Xbp1*, the following SYBR Green primers were used: *Xbp1(s):* Forward, CTGAGTCCGAATCAGGTGCAG; Reverse, GTCCATGGGAAGATGTTC TGG. *Xbp1(u):* Forward, TGGCCGGGTCTGCTGAGTCCG; Reverse, GTCCATGGGAAGATGTTCTGG.

## Determination of serum leptin and corticosterone levels by ELISA

Blood of random-fed mice was collected after decapitation, and serum was prepared by centrifugation at 5000xg and 4 °C for at least 30 min. Serum leptin levels were determined using the Mouse Leptin Enzyme-linked Immunosorbent Assay (ELISA) Kit (Crystal Chem, Cat# 90030) and serum corticosterone levels using the Corticosterone ELISA Kit (Enzo Life Sciences, Cat# ADI-900-979) according to the manufacturer's instructions. Serum samples of HFD-fed mice were diluted 1:40 (for leptin ELISA) or 1:30 (for corticosterone ELISA) with sample diluent prior to performing the assay. Optical density of each well was determined using a FilterMax F5 Multi-Mode microplate reader and SoftMax Pro 6.3 software (Molecular Devices). A four-parameter curve-fit standard curve was used for quantification.

## Isolation of neuronal and non-neuronal cell fractions from adult mouse brain

**Dissociation of adult mouse hypothalamus.** Adult mouse hypothalamus was dissociated using the Adult Brain Dissociation Kit for mouse and rat (Miltenyi Biotec, Cat#130-107-677) in combination with the gentleMACS Octo Dissociator with Heaters (Miltenyi Biotec, Cat# 130-096-427) according to the manufacturer's instructions. In brief, brains of three NCD random-fed control mice (Samples1-3, two females and one male; Supplementary Fig. 1g, i) at 40–43 weeks of age were resected and washed in ice-cold PBS. 2 mm coronal brain sections containing the hypothalamus were cut using a stainless-steel mouse brain matrix, from which the hypothalamus was dissected under a microscope (SteREO Discovery.V12, Zeiss). The hypothalamus was transferred to a gentleMACS C-tube (Miltenyi Biotec, Cat# 130-093-237) containing 1980 μL of enzyme mix and tissue dissociation was performed over 30 min at 37 °C using the 37C_ABDK_02 program operating the gentleMACS Dissociator. Afterward, samples were passed through a 70 μm MACS SmartStrainer (Miltenyi Biotec, Cat#130-098-462) and cells were pelleted by centrifugation for 10 min at 4 °C, followed by debris removal using a density gradient. To this end, cells were resuspended with 1550 μL cold DPBS (with Ca$^{2+}$, Mg$^{2+}$, 1 g/L D-glucose, and 36 mg/L sodium pyruvate (Gibco; Thermo Scientific, Cat# 14287080)), mixed with 450 μL cold Debris Removal Solution, and gently overlayed with 2 mL DPBS. Gradients were spun for 10 min at 3000xg and 4 °C with full acceleration and full break using a centrifuge (Heraeus Megafuge 16 R, Thermo Scientific) equipped with a TX-400 swinging bucket rotor (Thermo

Scientific). Of the three appearing phases, the upper two were discarded, the tube filled up with cold DPBS, and cells centrifuged at 4 °C and 1000xg for 10 min. Red blood cells were removed by incubating the cells with 500 μL cold 1x Red Blood Cell Removal Solution for 10 min at 4 °C. Subsequently, 5 mL of cold PB buffer (0.5% BSA (Miltenyi Biotec, Cat# 130-091-376) in DPBS) was added and cell suspensions centrifuged for 10 min at 300xg and 4 °C. Cells were then further processed to enrich the neuronal and non-neuronal cell fractions.

**Separation of neuronal and non-neuronal cell fractions.** Separation of neuronal and non-neuronal cell fractions was performed using the Neuron Isolation Kit for mouse (Miltenyi Biotec, Cat# 130-115-389) according to the manufacturer's instructions. This protocol is based on indirect magnetic labeling by using biotin-conjugated antibodies specific for non-neuronal cells (including astrocytes, oligodendrocytes, microglia, endothelial cells, and fibroblast) in combination with anti-Biotin MicroBeads. Cell pellets obtained from the dissociation of the mouse hypothalamus (see above) were resuspended in 100 μL PB buffer and incubated with 25 μL of a Non-Neuronal Cell Biotin-Antibody Cocktail for 5 min at 4 °C. Subsequently, cells were washed in PB buffer and 25 μL of Anti-Biotin MicroBeads were incubated with 100 μL of the cell suspension for 10 min at 4 °C. Afterward, 500 μL PB buffer were added and cell suspensions applied to equilibrated LS columns (Meltenyi Biotec, Cat# 130-042-401) attached to the magnetic field of a MACS separator (Miltenyi Biotec). The flow through containing unlabeled neurons was collected, and the LS columns washed another 6x with 1 mL PB buffer to collect residual cells. To collect the non-neuronal cell fraction, LS columns were removed from the separator and magnetically-labeled cells flushed out by pushing 3 mL PB buffer through the column. Separation of neuronal and non-neuronal cell fractions was verified using flow cytometry. An aliquot of cells (1:20) was stained using the antibodies anti-CD11b-BV421 (clone M1/70) (BioLegend, Cat# 101236, dilution 1:150), anti-ACSA-2-APC (clone IH3-18A3) (Miltenyi Biotec, Cat# 130-117-535, dilution 1:100) and anti-O4-APC (clone O4) (Miltenyi Biotec, Cat# 130-119-155, dilution 1:100) for 15 min at room temperature protected from light. Cells were washed twice with 1 mL FACS buffer (autoMACS Rinsing Solution supplemented with MACS BSA Stock Solution, Miltenyi Biotec) before being resuspended in 200 μL FACS buffer. Stained samples were analyzed immediately using MACSQuant 10 and software MACSQuantify v2.13.1 (Miltenyi Biotec). Dead cells were excluded based on propidium iodide fluorescence. Data were analyzed using FlowJo software (BD Biosciences).

## Assessment of mitochondrial morphology via Transmission Electron Microscopy (TEM)

**TEM analysis of mitochondrial morphology in N43/5 neurons.** N43/5 cells were seeded on Aclar fluoropolymer-film. The next day, cells were treated with the respective siRNA (scrambled or siCerS6) for 48 h followed by palmitate or BSA treatment for 9 h as described above. Subsequently, cells were fixed in pre-warmed fixative (2% glutaraldehyde, 2.5% sucrose, and 3 mM CaCl$_2$ in 100 mM HEPES, pH 7.4) for 30 min at RT and 30 min at 4 °C. Cells were rinsed 3x in 0.1 M cacodylate buffer (pH 7.2) and post-fixed in 1% OsO$_4$, 1.25% sucrose, 10 mg/mL potassium ferrocyanide in 0.1 M sodium cacodylate buffer for 1 h on ice. Following post-fixation, cells were rinsed in ddH$_2$0, dehydrated through an ethanol series, and embedded in epoxy resin. Ultrathin sections (70 nm) were cut with a diamond knife (Diatome) on an ultramicrotome (EM-UC6, Leica) and contrasted with 1.2% uranyl acetate and lead citrate (Reynolds solution). Images were taken on a transmission electron microscope (JEOL JEM 2100Plus), camera OneView 4 K 16 bit (Gatan), and software DigitalMicrograph (Gatan, version 3.32.2403.0) at 80 kV at RT. Images were analyzed using the ImageJ (Fiji)-software package (version

2.9.0/1.53p13). Mitochondria were manually traced and measured for aspect ratio.

**TEM analysis of mitochondrial morphology in mouse POMC neurons.** CerS6$^{ΔPOMC}$ and littermate control mice at 13 weeks of age that had received a HFD for 8 weeks, along with control mice that received a NCD, were deeply anesthetized and transcardially perfused first with ice-cold PBS followed by perfusion with a freshly prepared picric acid-paraformaldehyde-glutaraldehyde fixative (4% PFA, 15% picric acid, and 0.08% glutaraldehyde in PBS, pH 7.4). After post-fixation overnight at 4 °C in the same fixative, brains were transferred to PBS. Brain sections (50 μm) were cut on a vibratome and sections containing the ARC were immunostained with anti-POMC primary antibody (dilution 1:7500; Phoenix Pharmaceuticals, Cat# H-029-30). After overnight incubation at room temperature, sections were washed with PBS, incubated with biotin-conjugated donkey anti-rabbit IgG secondary antibody (dilution 1:250; Jackson ImmunoResearch Laboratories, Cat# 711-065-152) for 2 h, washed again, put in avidin-biotin complex (ABC; Vector Laboratories), and developed with 3,3-diaminobenzidine (DAB). Sections were osmicated (15 min in 1% osmium tetroxide) and dehydrated in increasing ethanol concentrations. During the dehydration, 1% uranyl acetate was added to the 70% ethanol to enhance ultrastructural membrane contrast. Flat embedding in Durcupan followed dehydration. Ultrathin sections were cut on a Leica ultramicrotome, collected on Formvar-coated single-slot grids, and analyzed with a Tecnai 12 Biotwin electron microscope (FEI) with an AMT XR-16 camera. Hypothalamic sections containing POMC immunoreactive cells were analyzed by electron microscopy. The mitochondrial aspect ratio was calculated using ImageJ.

## Sphingolipidomics

**Sphingolipid analysis in N43/5 neurons and hypothalamus samples.** Levels of selected sphingolipid species in N43/5 cells and mouse hypothalamus were determined by Liquid Chromatography coupled to Electrospray Ionization Tandem Mass Spectrometry (LC-ESI-MS/MS) using previously described procedures:[84–86] Approximately $3 × 10^6$ N43/5 cells were homogenized in 300 μL water using the Precellys 24 Homogenizer (Peqlab) at 6500 rpm for 30 s. The protein content of the homogenate was routinely determined using bicinchoninic acid. To 100 μL homogenate we added 500 μL methanol, 250 μL chloroform, and internal standards (127 pmol Ceramide 12:0, 130 pmol Glucosylceramide 12:0, and 124 pmol Sphingomyelin 12:0; Avanti Polar Lipids). For sphingolipid analyses in mouse hypothalamus (Supplementary Fig. 3d-f), tissue was homogenized in 100 μL water. 20 μL of the homogenate were diluted to 100 μL with water and further treated as described above. Alternatively (Fig. 1d, Supplementary Fig. 1e, f), 50 μL of hypothalamus lysates prepared in lysis buffer (50 mM Tris, 130 mM NaCl, 5 mM EDTA, 1% NP-40, 1 mM PMSF, 1 mM NaF, 1x Halt protease and phosphatase inhibitor cocktail (Thermo Scientific, Cat# 78446)), containing 100 μg of protein, were diluted to 100 μL and used for lipid extraction. Lipid extraction was performed as previously described.[84,85] After adding the internal standards, 500 μL of methanol, and 250 μL of chloroform, the mixture was sonicated for 5 min and lipids were extracted overnight at 48 °C. Interfering glycerolipids were degraded by alkaline hydrolysis by adding 75 μL of 1 M potassium hydroxide in methanol. After 5 min of sonication, the extract was incubated for 2 h at 37 °C and then neutralized with 6 μL of glacial acetic acid. 2 mL of chloroform and 4 mL of water were added to the extract, which was vortexed vigorously for 30 s and then centrifuged (4000xg, 5 min, 4 °C) to separate layers. The lower (organic) phase was transferred to a new tube and the upper phase extracted with 2 mL of chloroform. The combined organic phases were dried under a stream of nitrogen. Dried lipid extracts were resolved in 300 μL mobile phase solvent A (see below). LC-MS/MS analysis was performed using a normal phase Nucleosil NH2 column (50 mm × 2 mm ID, 3 μm particle size,

120 Å pore size, Macherey-Nagel) with detection using a QTRAP 6500 mass spectrometer operated by Analyst 1.6.3 (SCIEX). The LC (1260 Infinity Binary LC System, Agilent) was operated at a flow rate of 0.75 mL/min with a mobile phase of acetonitrile/methanol/acetic acid 97:2:1 (v/v/v), with 5 mM ammonium acetate (solvent A) and methanol/acetic acid 99:1 (v/v) with 5 mM ammonium acetate (solvent B). LC separation and MS/MS detection were performed as previously described:[84,85] Prior to sample injection, the column was equilibrated for 1.0 min with 100% A. After injection of 20 μL of the sample, 100% A was continued for 0.5 min, then linearly changed to 90% A/10% B in 0.2 min and held for 0.5 min, then linearly changed to 82% A/18% B in 0.4 min and held for 0.6 min, followed by a 0.4 min-linear gradient to 100% B, which was maintained for 1.9 min and finally restored to 100% A by a 0.4 min-linear gradient and held for 1.6 min to re-equilibrate the column. The total run time was 6.5 min. Sphingolipid species were monitored in the positive ion mode with their specific Multiple Reaction Monitoring (MRM) transitions[84–86]. The instrument settings for nebulizer gas (Gas 1), turbogas (Gas 2), curtain gas, and collision gas were 50 psi, 55 psi, 20 psi, and medium, respectively. The interface heater was on, the Turbo V ESI source temperature was 400 °C, and the ionspray voltage was 5.5 kV. For all MRM transitions the values for declustering potential, entrance potential, and cell exit potential were 80 V, 10 V, and 14 V, respectively. The collision energies ranged from 35 to 55 V. Sphingolipid species were quantified on the basis of calibration curves calculated from LC-MS/MS measurements of serially diluted synthetic sphingolipid standards (Avanti Polar Lipids). Data analysis was performed with MultiQuant 3.0.3 (SCIEX). The calculated amounts of sphingolipids were normalized to the protein content (N43/5 cells and hypothalamus lysates) or, if sample was limited, the wet weight (hypothalamus tissue) of the respective sample.

**Sphingolipid analysis in neuronal and non-neuronal cell fractions.** Lipidomic analysis in neuronal and non-neuronal cell fractions was performed as described previously:[87] Extraction was performed by mixing cell pellets with extraction buffer consisting of methanol/methyl *tert*-butyl ether/chloroform 4:3:3 (v/v/v) and 100 pmol/sample internal standard (Ceramide d18:1/12:0) for 1 h at 37 °C using the Eppendorf ThermoMixer. Afterward, the single-phase supernatant was collected, dried under $N_2$, and stored at −20 °C. Before analysis, lipids were dissolved in MeOH and separated on a C30 LC column using a gradient elution with (A) acetonitrile/water (6:4) with 10 mM ammonium acetate and 0.1% formic acid, and (B) isopropanol/acetonitrile (9:1) with 10 mM ammonium acetate and 0.1% formic acid at a flow rate of 260 μL/min. Eluted lipids were analyzed on a Q-Exactive HRMS operated by Xcalibur software (Thermo Scientific) in positive and negative mode using heated electrospray ionization (HESI). Lipids were identified by predicted mass (resolution 5 ppm), retention time and specific fragmentation patterns. Data analysis was performed with TraceFinder 5.1 (Thermo Scientific).

**In situ hybridization (BaseScope)**
Random-fed adult mice were deeply anesthetized and transcardially perfused first with ice-cold PBS followed by 4% (w/v) paraformaldehyde (PFA in PBS, pH 7.4). Brains were post-fixed in 4% PFA in PBS at 4 °C overnight and lowered in 20% (w/v) sucrose in PBS. Brains were cut at 20 μm on a sliding microtome (Leica Microsystems, SM2010R). Sections were mounted on SuperFrost Plus GOLD glass slides (Menzel-Gläser), air-dried, and frozen at -80 °C until further processed. In situ hybridization was performed according to the manufacturer's protocol for BaseScope Kit v2-RED (Advanced Cell Diagnostics (ACD), Cat# 323900; Supplementary Fig. 3 h, 4c) or Duplex Detection Kit (ACD, Cat# 323800; Supplementary Fig. 5b, 6b, 7b). All BaseScope reagents were purchased from ACD unless otherwise stated. On the day of the assay, sections were dried at 60 °C for 4–6 h, submerged in Target Retrieval reagent (Cat# 322000) at 98-100 °C for 7 min using a steam

cooker, rinsed in autoclaved dd$H_2O$, and dehydrated in 100% EtOH. The following day, sections were incubated with Protease III (Cat# 322340) for 30 min at 40 °C. Hybridization of specific probes (see below), amplification, and detection steps were performed according to the manufacturer's protocol. Sections were counterstained with Haematoxylin and cover-slipped with VectaMount permanent mounting media (Vector Laboratories Inc.). Brightfield images were acquired at 20x or 40x magnification on an Axio Imager 2 microscope operated by ZEN 2 (blue edition) software using an Axiocam 105 color camera (Carl Zeiss Microscopy GmbH). Separate images of one brain slice were taken and aligned using the "Pairwise stitching" plugin in Fiji[88]. Standard image processing was performed (e.g., adjustment of brightness and contrast) using Microsoft PowerPoint.
The following probes have been used for in situ hybridization:

- BA-Mm-Cers6-E3E5 (Cat# 703401); Accession No: NM_172856.3, target region [bp]: 455-555
- BA-Mm-Cers6-E4E5 (Cat# 703411); Accession No: NM_172856.3, target region [bp]: 511-555
- BA-Mm-Cers1-E1E3 (Cat# 703381); Accession No: NM_138647.3, target region [bp]: 307-505
- BA-Mm-Cers1-E2E3 (Cat# 703391); Accession No: NM_138647.3, target region [bp]: 465-507
- BA-Mm-AgRP-E3E4-C2 (Cat# 875741-C2); Accession No: NM_001271806.1, target region [bp]: 419-462
- BA-Mm-Pomc-E2E3-C2 (Cat# 875751-C2); Accession No: NM_008895.4, target region [bp]: 260-297
- BA-Mm-Nr5a1-E3E4-C2 (Cat# 1000091-C2); Accession No: NM_139051.3, target region [bp]: 420-454

**Analysis of leptin-stimulated pSTAT3 immunoreactivity in POMC neurons**
To assess leptin sensitivity in POMC neurons in the ARC, the immunoreactivity of pSTAT3 in POMC-expressing cells was assessed after administering IP injections of recombinant murine leptin (6 mg/kg body weight; PeproTech; Thermo Scientific, Cat# AF-450-31) dissolved in PBS to 23 weeks old *CerS6*^ΔPOMC mice and their control littermates that had received the HFD for 20 weeks. Prior to the experiment, mice underwent a 16-h overnight fasting period. Mice were decapitated 45 min after leptin injections. Brains were promptly resected, placed in an OCT embedding matrix, and frozen in cold isopentane with liquid nitrogen. Subsequently, brain sections measuring 20 μm in thickness were obtained using a sliding microtome, mounted on SuperFrost glass slides (Menzel-Gläser), and subjected to further processing for POMC and pSTAT3 immunohistochemistry using the TSA Plus Cyanine 3 Kit (Akoya Biosciences, Cat# NEL744001KT). To maintain uniform immunostaining, sections were processed simultaneously under identical conditions. Sections were allowed to air-dry for 5 min at room temperature and post-fixed in 2% PFA for 45–60 min, incubated with 0.3% $H_2O_2$ for 15 min, and treated with ice-cold MeOH for 5 min. Between each step, slides were washed twice for 10 min in PBS. Sections were then incubated for 1 h in TNB blocking buffer (0.1 M TRIS-HCl, pH 7.5; 0.15 M NaCl; 0.5% blocking reagent (Akoya Biosciences, Cat# FP1020)), and for 48 h at 4 °C in primary antibody targeting pSTAT3^Tyr705 (phospho-STAT3 Tyr705 (D3A7) XP®, Cell Signaling, Cat# 9145; 1:100 diluted in TNB blocking buffer). Thereafter, sections were washed in PBS with 0.3% Triton X (3 × 10 min) and incubated for 30 min with secondary antibody (anti-rabbit IgG (goat) HRP, MAb Technologies, Cat# NEF812001EA; 1:100 diluted in PBS with 0.2% Triton X). Slides were washed again and incubated in TSA Plus Cyanine 3 (Akoya Biosciences, Cat# TS-000202; 1:100 dilution in amplification buffer) for 10 min. Afterward, sections were washed 3 × 10 min with PBS/0.3% Triton X, quickly dipped in 4% PFA, and washed 3 × 10 min in PBS. Sections were incubated for 60 min in blocking solution (3% donkey serum in PBS/0.3% Triton X), directly followed by incubation with an antibody targeting POMC (Phoenix Pharmaceuticals, Cat# H-029-30),

diluted 1:1000 in SignalStain Ab Diluent (Cell Signaling, Cat# 8112), for 48 h at 4 °C. Afterward, slices were washed 3 × 10 min with PBS/0.3% Triton X and incubated for 60 min with the secondary antibody (anti-rabbit IgG, Alexa 488; Invitrogen, Cat# A21206; diluted 1:1000 in PBS with 0.2% Triton X) containing Hoechst 33342 (1:1000) for nuclear staining (Invitrogen, Cat# H1399). Sections were mounted with Vectashield (Vector Laboratories Inc., Cat# H1000). Images of the ARC were captured at the median eminence area from three brain slices per mouse using a Leica TCS SP-8-X confocal microscope equipped with x40/1.30 oil objective. Tile scans were automatically assembled by the LasX software. Same microscope settings for the detection of pSTAT3 (including objective, zoom, laser power, gain) were used to acquire all pictures. For quantification and image representation, maximum intensity projections of z-stacks were produced using the Fiji software package.

**Quantification.** Relative amounts of pSTAT3-positive cells were determined using custom-made, semi-automatic ImageJ macros. For total amounts of pSTAT3-positive cells in the ARC, DAPI-positive and pSTAT3-positive cells were detected automatically. Relative values were expressed by ratio of pSTAT3-positive cells to the total number of DAPI-positive cells. For the assessment of relative numbers of pSTAT3-positive POMC neurons, regions of interest (ROIs) corresponding to POMC neurons were drawn manually as POMC neurons could not be detected automatically in a reliable way due to high inter-image variability in the intensity of the POMC signal. Subsequently, POMC ROIs were binarized and used to mask the pSTAT3-channel. Thereby, the following automatic detection of pSTAT3-positive cells could be limited to POMC neurons. In both analyses, quality checks were included in the pipeline in which the user was given the chance to supervise the macro's performance, and correct potential errors manually. All parameters for the required preprocessing steps were held constant throughout the analysis, and the analyst was blinded to the conditions.

**Mapping CerS expression in the hypothalamus of mice using HypoMap**
The functions DotPlot and FeaturePlot from the Seurat package (https://satijalab.org/seurat/; version: 4.3.0) were used to visualize the expression of the *CerS* genes in the previously published HypoMap dataset (available from: https://www.repository.cam.ac.uk/handle/1810/340518)[32].

**Statistical analysis**
Data are generally shown as mean values ± standard error of the mean (SEM) or as boxplots, including data points of individual mice entering the analysis. Sample numbers of data obtained from mice refer to the number of individual mice entering the analysis as specified in the figure legends; the first n-value refers to the control group and the second n-value to the test group. Experiments on cultured cells were performed in at least three independent experiments with one or more technical replicates per experiment as also specified in the figure legends. Comparison of two independent groups was performed with unpaired two-tailed Student's *t*-test. One-way ANOVA followed by Tukey's post hoc multiple comparison test was performed to compare one variable between three groups. For statistical analyses of groups comparing two variables (e.g., genotype and treatment), a two-way ANOVA was conducted, followed by Tukey's or Bonferroni's post hoc multiple comparison test, as specified in the figure legend. For statistical analysis of groups comparing the three variables siRNA treatment, BSA/PAL exposure, and time, a three-way ANOVA was conducted, followed by Tukey's post hoc multiple comparison test. Longitudinal data comparing two groups was analyzed using two-way repeated-measures (RM) ANOVA. Results of the two-way RM ANOVA are reported for the variable genotype and for the statistical interaction of

the variables genotype and time as specified in the respective graphs. When body weight data of mice was not recorded weekly leading to missing data points in the body weight curve, a mixed effects model (RMEL) was used. Probability plots were utilized to estimate changes in mitochondrial aspect ratio and area in POMC neurons of mice and statistical differences were tested using the Kolmogorov-Smirnov test. Energy expenditure over body weight (Fig. 3i) was analyzed by analysis of covariance (ANCOVA) using CalR Version 1.3[89]. All figures were generated and all other statistical analyses were performed using the GraphPad Prism 9 software. Details of statistical analyses are specified in the Source Data file.

**Reporting summary**
Further information on research design is available in the Nature Portfolio Reporting Summary linked to this article.

## Data availability
Source data are provided with this paper. Uncropped immunoblots are presented in the Source Data file. The seurat object containing HypoMap, which is required to reproduce the single-cell data related figures is available at University of Cambridge's Apollo Repository (https://doi.org/10.17863/CAM.87955). Source data are provided with this paper.

## Code availability
The custom made semi-automatic ImageJ macros used to quantify pSTAT3 signal intensity from immunohistochemical stainings are available under https://github.com/mrfeldmann/ceramide_paper. The R code used to analyze HypoMap single cell sequencing data is available under https://github.com/lsteuernagel/ceramide_paper_hypomap.

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

## Acknowledgements

We wish to thank Änne Lautenschläger, Nadine Spenrath, Pia Scholl, Christian Heilinger, Alexandra Scharn, Nadine Zgoda, and Ashley Kristant for technical assistance. We thank Jens Alber for performing indirect calorimetry measurements and the analysis of body composition from CT images. We thank Dr. Astrid Schauss and her team for support with TEM imaging of N43/5 cells. Illustrations of mice were generated using Biorender.com. This work was supported by the European Research Council (ERC) under the European Union's Horizon 2020 research and innovation program (grant agreement No 742106) to J.C.B.; by National Institute of Health grants DK126447 and AG067329 and a Klarman Family Foundation grant to T.L.H. T.H. received support by the Swiss National Science Foundation (SNF 31003A_179371) and under the framework of the European Joint Program on Rare Diseases (EJP RD + SNF 32ER30_187505). This work was in part supported by the CECAD (funded by the DFG within the Excellence Initiative by German Federal and State Governments) and funds by the DZD and the CMMC (to J.C.B.). This work is funded by the Deutsche Forschungsgemeinschaft (DFG, German Research Foundation) - SFB 1218 - Projektnummer 269925409 (to J.C.B.).

## Author contributions

P.H., S.M.S, and J.C.B. conceived the present study. P.H. and S.M.S. planned, supervised, and executed most of the experiments and analyzed most of the data. S.M.S conducted metabolic characterization of CerS1ΔNkx2.1 and CerS6ΔNkx2.1 mice. P.H. performed metabolic characterization of CerS6ΔPOMC, CerS6ΔAgRP, and CerS6ΔSF-1 mice, analysis of CD versus HFD-fed mice, isolation of neurons and non-neuronal cell fractions, and most the experiments on cultured cells. M.Q.C. and R.F.U.K. supported cell culture experiments. C.L.B. performed Western blot analysis of N43/5 cells. M.Q.C. performed BaseScope analysis on CerS6ΔNkx2.1 brains. L.S. analyzed *CerS* expression using HypoMap. A.d.R.-M. analyzed mitochondrial respiration in N43/5 cells using Seahorse. C.A.B. resected hypothalami of CD and HFD-fed C57BL/6 N mice. M.F. wrote and ran the ImageJ script for automated quantification of pSTAT3 immunoreactivity from confocal images. E.K. maintained the HFD-fed CerS6Δ/Δ cohort. V.K. supported the flow cytometry-based analysis of hypothalamic cell fractions. L.V. and T.L.H. conducted TEM imaging of POMC neurons and quantification of mitochondrial shape parameters. A.M. and T.H. performed lipidomic analysis on isolated neurons and non-neuronal cell fractions. S.B. performed all

other lipidomic analyses. M.P., N.B. and A.S. helped with additional experiments. P.H. wrote the manuscript with input from the co-authors.

## Funding

## Competing interests
J.C.B. is co-founder of Cerapeutix and has received research funding through collaborations with Sanofi Aventis and Novo Nordisk Inc., which did not affect the content of this article. The remaining authors declare no competing interests.

## Additional information

[1]Department of Neuronal Control of Metabolism, Max Planck Institute for Metabolism Research, Gleueler Strasse 50, 50931 Cologne, Germany. [2]Policlinic for Endocrinology, Diabetes and Preventive Medicine (PEDP), University Hospital Cologne, Kerpener Strasse 26, 50924 Cologne, Germany. [3]Cologne Excellence Cluster on Cellular Stress Responses in Aging-Associated Diseases (CECAD) and Center for Molecular Medicine Cologne (CMMC), University of Cologne, Cologne, Germany. [4]Max Planck Institute for Metabolism Research, Research Group Neurocircuit Wiring and Function, Cologne, Germany. [5]National Center for Diabetes Research (DZD), Ingolstädter Landstrasse 1, 85764 Neuherberg, Germany. [6]Faculty of Mathematics and Natural Sciences, University of Cologne, Cologne, Germany. [7]Yale Center for Molecular and Systems Metabolism, Department of Comparative Medicine, Yale University School of Medicine, 310 Cedar St., BML 330, New Haven, CT 06520, USA. [8]Laboratory of Glia-Neuron Interactions in the Control of Hunger. Achucarro Basque Center for Neuroscience, Leioa 48940, Spain. [9]Ikerbasque-Basque Foundation for Science, Bilbao 48013, Spain. [10]Center for Integrative Human Physiology, University of Zürich, Zürich, Switzerland. [11]Institute of Clinical Chemistry, University Hospital, Zürich, Switzerland. ✉e-mail: bruening@sf.mpg.de

