## [Peer Review File · Nature Communications]

CerS6-dependent ceramide synthesis in hypothalamic neurons promotes ER/mitochondrial stress and impairs glucose homeostasis in obese miceReviewers' comments:

Reviewer #1 (Remarks to the Author):

Summary

Hammerschmidt et al. report that high fat diets increase CerS6 expression in the hypothalamus of mice. They then show that the deletion of CerS6 in various hypothalamic nuclei can lead to improved insulin and glucose regulation and, in some cases, reduced adiposity. The authors conclude that the accumulation of C16 ceramides in the hypothalamus results in ER/mitochondrial stress, leading to alterations in appetite. The results provide potentially important insight into our understanding of the mechanisms by which increased fatty acids contribute to hypothalamic neuron dysfunction in the over-fed state.

While the study includes important findings, it lacks some crucial experiments that could improve the results and their interpretation.

Comments

The authors report that they achieved hypothalamic neuron-specific inactivation of *Cers6* using the *Nkx2.1* promoter (for Cre expression). However, the *Nkx2.1* promoter can affect many regions of the brain, including the cortex and hippocampus. The authors make the case that they are probing both by *in situ* hybridization and qPCR for mRNA lacking exon 4; however, they do not provide the probe sequences or validation that they are specific to an altered form of *Cers6*. In the mice for Figure 2, the authors show an overall reduction of the full *Cers6* expression in various parts of the brain. If the full transcript of *Cers6* is reduced only in the hypothalamus, the authors can make the case that the *Nkx2.1* Cre expression is hypothalamic-specific. In Figure 3, the authors should show that the full *Cers6* transcript is reduced in the VMH dissected out of the hypothalamus. In Figures 4 and 5, the authors should show the same with the ARC dissected from the hypothalamus.

The authors nicely show *Cers6* protein blots from the whole hypothalamus in Figure 1. These blots should be present for the knockout mice throughout the paper to further show

that CerS6 is inhibited/reduced in the knockout animals. In addition, various brain structures such as the hippocampus and cortex should be included in these western blots for the Nkx2.1 Cre animals.

In Figures 2-5, the authors only superficially mention the possibility that CerS6 deletion could affect development. In Figure 2, they make the case that rodent length is unchanged and use this as proof that there are no developmental effects on the mice; while it may be true that there are no growth effects, body length does not account for the possibility of developmental issues with neural circuitry that can affect feeding and obesity regulation throughout/later in life. A discussion of the possible developmental caveats of the Nkx2.1 and Pomc Cre mice should be included in this manuscript.

The authors state for Figure 2e, and other following figures, that reduced adiposity is consistent with reduced circulating leptin levels. While it is possible that reduced adiposity may result in less leptin production, a reduction in leptin should result in increased eating behavior and increased obesity/adiposity. These statements need to be clarified in the text. Leptin tolerance tests are also highly suggested to determine if leptin signaling is altered. Figure 5 should include circulating leptin, and for these Pomc-Cre mediated knockout of CerS6 mice, a leptin tolerance test is critical to know if these neurons are functioning correctly.

Figure 2 h-j and subsequent figures, “genotype” and “interaction” need to be defined in the figure legend so that the reader understands what the two p values represent.

Extended Data Figure 5 shows little association of CerS6 transcript with Pomc transcript. However, the variant CerS6 transcript does associate more closely. Better images or quantification of the loss of CerS6 need to be shown. In addition, neurons from other nuclei within the hypothalamus that arise from Pomc expressing founder cells during development should also be quantified for CerS6 loss.

In the results section for Figure 6, the authors need to explain why Xbp1 splicing is important.

It has been shown that leptin signaling is altered in Pomc neurons with increasing glucosylceramides. Therefore, figure 6 would be further enhanced by adding glucosylceramide data.

The mHypoE-N43/5 cells used for Figure 6 are derived from female mice. All of the data presented in this manuscript for Agrp and Pomc Cre animals are from male mice, and the SF1 Cre mediated deletion CerS6 led to sexually dimorphic results. The study would be dramatically enhanced using primary neurons taken from the knockout mice generated by the lab.

Figure 6 makes a good case for CerS6 and C16 ceramide being deleterious for hypothalamic neurons. However, it makes a better Figure 1 introducing why the authors studied mice with a CerS6 knockout rather than correlating the data as a possible mechanism. This lab has already established the importance of mitochondrial C16 ceramides; showing it in hypothalamic neurons is important, but the study would be significantly enhanced by showing mitochondrial dysfunction in vivo, not in vitro.

Sphingolipid data (concentrations and species) needs to be done in the hypothalamus from the mouse lines generated. Ideally, sphingolipids need to be obtained from the specific nuclei in which the CerS6 knockout occurs. For the Nkx2.1 Cre knockout mice, sphingolipid data should come from various regions of the brain. This is necessary to show that the knockout mice are functioning as expected and that exogenous sources of C16 ceramides aren't incorporated into the various regions/hypothalamic nuclei.

The ER stress and mitochondrial phenotype seen in Figure 6 should be shown in the hypothalamus or at least in primary cultured neurons from the knockout mice. In addition, ER/mitochondrial stress markers such as those used in Figure 1 should be assessed in the knockout mice.

While it is easy to assume ER/mitochondrial stress would lead to neuron dysfunction, there is no evidence in this study that that happens when C16 ceramides are elevated in neurons. Functional data should be included.

Line 382, extend should be extent.

Reviewer #2 (Remarks to the Author):

In the manuscript by Hammerschmidt et al, the authors investigated the role of CerS6-dependent ceramide synthesis in the hypothalamus and its role in regulating insulin resistance in murine models of obesity. The relevance of this work is high, as it adds details to the possible mechanism through which ceramides could control appetite and regulate metabolism in nutritional studies, suggesting that CerS6-generated ceramides are the main regulators and that this enzyme could be a possible therapeutic target.

The authors made an important effort in generating several models to test their hypothesis. However, I am not convinced that the main conclusion, that Cer 16:0 is the main molecular messenger in the described model, can be fully supported without having measured ceramides in most of the reported results. I believe the current version of this manuscript should be significantly improved by the authors and I hope the following comments might clarify my conclusions.

More in detail:

- In the introduction, a description of the specificity of each Ceramide synthase should be provided, to help readers understanding what is the possible overlap between these enzymes in terms of generated products; for example, CerS5 and CerS6 can both generate C16 ceramides. This I think will be a very useful information to include.
- This work is based on the elimination of CerS6 activity to significantly decrease the level of C16 ceramides. However, the product and the other ceramides level, that could be affected by the ablation of CerS6, are never measured in the target cells or in the whole hypothalamus. I think these data would be an important confirmation to support the authors findings and claims. But without measuring them, how can the authors be sure that CerS5 for example is not compensating for the lack of CerS6 activity? Was the measurement attempted other than in the invitro example? Please explain.
- What is the meaning of the first sentence of the Summary "Ceramide form a heterogeneous group of acyl chain length-specific sphingolipids..."? considering that almost all lipids are.
- Abbreviations are used in the Summary before explanation of their meaning, please amend.
- Introduction lines 79-83: please explain a bit more in detail, by introducing a couple of examples of the findings.

- Lines 84-85: ceramides are not only differing by their acyl chain lengths but also by saturation state and long chain base type. Please amend to give a full picture of their heterogeneity.
- Line 97: please specify which inhibitor the authors are referring to.
- Lines 134-135: how was ATF4 selected? As it is the only gene mentioned here.
- Lines 137-39: in which mouse model was this done?
- I understand that CerS5 might have been excluded as a possible target due to previous analysis made by the authors and reported in literature. But its distribution in the hypothalamus is very similar to CerS6. Why ablation of CerS5 was not considered as a control here? For example, in the paragraph starting at line 198 the authors describe the inhibition of CerS1, which is a good control. But CerS5 could have been added as well.
- Lines 242-266: the authors describe an interesting sex-specific effect. Was this observed/excluded in the experiments reported in the previous paragraphs? A more detailed discussion on sex dimorphism reported earlier in similar conditions would be useful to the Discussion section.
- Lines 345-350: a similar effect can be observed for C24:0 Ceramides, any comment on this? As CerS6 is also generating C14 ceramides, and these are showing a similar profile as C16, why the author excluded the possibility of C14 being relevant in this model?
- Line 375-85: this is a possible conclusion, although CerS can also generate deoxy-ceramides that have been shown to induce mitochondrial fragmentation (Alecú et al, 2017 JLR). Was that possibility considered?
- More in general: CerS directly generate dehydro-ceramides but the authors only mentioned ceramides in the manuscript. Was the possibility of dehydro forms being the main player in this model considered? Please comment on this, especially as the authors always refer to de novo synthesis of sphingolipids.
- Lines 405-408: the link between infusion of C6 ceramides and effect onto C16 and 18 ceramides level is not clear. Please explain.
- Lines 408-413: this is a possible explanation, although, without measuring all the ceramides, we cannot exclude that for example accumulation of longer species might play a role too.
- Line 466: "...critical role of a CerS6-derived ceramide/ER stress axis..." it would be extremely interesting to estimate the fold change in ceramide levels required to trigger this

response. Was this done?

- Line 469: what is the example of C6 ceramide infusion telling us? Is it not going against the existence of acyl chain-specific effect of your model? Please explain.

- The lipidomic methodology is not described in the method section. I understand it was previously reported in another publication but most of the procedure should also be reported here to help readers.

Reviewer #3 (Remarks to the Author):

This manuscript by Hammerschmidt et al proposes new cell type-specific mechanisms through which ceramides, and their synthetic enzymes, regulate metabolism. The studies presented here use a nice combination of molecular genetics, behavior, physiology, and in vitro molecular studies to demonstrate the role of ceramide synthase 6 (CerS6) in regulating metabolic outcomes in diet-induced obese mice. The work presented here is novel, appropriate in scope for the journal, and will provide a nice resource for understanding the cell-selective effects of ceramides in the hypothalamus. Below are some comments that might help to clarify some of the findings from this work.

Major comments:

Diverging effects of cell type specific KO's – how do the authors explain the divergence of effects for the near-whole hypothalamus CerS6 KO (in the Nkx2.1-Cre line) versus the cell subtypes (in this case, where effects were shown – SF1 and POMC)? For example, there is a strong effect on insulin sensitivity in the SF1 cKO, but not in the Nk2.1 cKO. Additionally, the GTTs at 6 and 16 hr do not accord across the different cKO lines.

Magnitude of effect of metabolic state on CerS6 expression – CerS6 in the hypothalamus is significantly, but only modestly, increased in expression at the level of mRNA in db/db, HFD (although the dispersion of CerS6 protein expression is much broader); whereas, some of the metabolic phenotypes are more robust. What would account for this? Is this simply due to cell-selective changes of CerS6 expression across different hypothalamic subtypes? Or do the authors think that there may also exist other adaptations, such as post-translational

modifications of CerS6 that could account for the observed phenotypes?

Minor comments:

Fig. 1 – can the authors comment further on the (marked) increase in CerS5, relative to CerS6 (or all other CerS enzymes in HFD, for that matter)? Might their hypothesized differential control of subcellular ceramide pools lead to distinct metabolic outcomes (as suggested in Hammerschmidt et al., Cell 2019)? In other words, are CerS5/6 gene expression changes hypothesized to act in an oppositional fashion? Also, does CerS5 expression correlate with body weight, as well?

Fig. 2 – the CerS6 cKO in Nkx2.1-Cre leads to a small, but significant decrease (relative to controls) in adiposity and body weight, when the mice are placed on HFD. However, there is no observed change in energy intake or expenditure. What do the authors think is happening here?

Fig. S1 – addition of dot plots in the scRNA-seq analysis would be useful to compare expression of the CerS isoforms across populations (i.e. SF1, Agrp, Pomc).

Reviewer #1:

Summary

Hammerschmidt et al. report that high fat diets increase CerS6 expression in the hypothalamus of mice. They then show that the deletion of CerS6 in v hypothalamic nuclei can lead to improved insulin and glucose regulation & some cases, reduced adiposity. The authors conclude that the accumulation of C16 ceramides in the hypothalamus results in ER/mitochondrial stress, leading to alterations in appetite. The results provide potentially important insight into our understanding of the mechanisms by which increased fatty acids contribute

to hypothalamic neuron dysfunction in the over-fed state. While the study includes important findings, it lacks some crucial experiments that could improve the results and their interpretation.

We thank the reviewer for acknowledging the importance of our work and its relevance to the field. We appreciate the thorough review of our manuscript and the constructive criticism. We believe that we were able to significantly improve the content of the manuscript by answering the reviewer's questions and hope to have adequately addressed his/her concerns.

Comments

The authors report that they achieved hypothalamic neuron-specific inactivation of *Cers6* using the *Nkx2.1* promoter (for Cre expression). However, the *Nkx2.1* promoter can affect many regions of the brain, including the cortex and hippocampus. The authors make the case that they are probing both by in situ hybridization and qPCR for mRNA lacking exon 4; however, they do not provide the probe sequences or validation that they are specific to an altered form of CerS6.

We thank the reviewer for bringing to our attention that we had omitted this important information in the initial version of the manuscript. We have now included the necessary information on the probes used for BaseScope in situ hybridization in the Methods section of the revised version of the manuscript.

In the mice for Figure 2, the authors show an overall reduction of the full CerS6 expression in various parts of the brain. If the full transcript of *Cers6* is reduced only in the hypothalamus, the authors can make the case that the *Nkx2.1* Cre expression is hypothalamic-specific.

We thank the reviewer for this comment and apologize if we haven't been clear in properly explaining our results in this case. By qPCR analysis and BaseScope in situ hybridization probing against exon 4 of the CerS6 mRNA (the one loxP flanked in the modified gene) we show a reduction of CerS6 expression specifically in the hypothalamus in CerS6^{ΔNkx2.1} mice (i.e., the mice for Figure 2 of the initial version and Figure 3 of the revised version of the manuscript), but

not in other brain regions. We have revised the text accordingly and hope to have now formulated this finding more clearly.

In Figure 3, the authors should show that the full CerS6 transcript is reduced in the VMH dissected out of the hypothalamus. In Figures 4 and 5, the authors should show the same with the ARC dissected from the hypothalamus.

Due to the limited number of POMC/AgRP neurons in the ARC and SF-1 neurons in the VMH of the hypothalamus, validation of a reduction in CerS6 expression in the respective conditional knockout lines (CerS6^{ΔAgRP}, CerS6^{ΔPOMC}, CerS6^{ΔSF-1}) was not possible by qPCR. To still address the reviewers concern, we have performed PCR analysis on genomic DNA extracted from the respective hypothalamic nuclei, conforming the site-specific Cre-mediated recombination of the floxed CerS6 alleles in all mouse lines. In addition, we have performed BaseScope in situ hybridization using probes targeting either the full-length transcript of CerS6, revealing site-specific reduction in CerS6 expression in the conditional knockout mice, or the modified exon 4-deficient transcript of CerS6, revealing expression of the recombined mRNA of CerS6 specifically in the targeted nuclei of the hypothalamus. In none of the mouse models we observed recombination of CerS6 in a significant proportion of cells of the hippocampus or other brain areas except for the specifically targeted hypothalamic nuclei. These extensive new data are provided in the revised version of the manuscript.

The authors nicely show CerS6 protein blots from the whole hypothalamus in Figure 1. These blots should be present for the knockout mice throughout the paper to further show that CerS6 is inhibited/reduced in the knockout animals. In addition, various brain structures such as the hippocampus and cortex should be included in these western blots for the Nkx2.1 Cre animals.

We have included Western blots for CerS6 in hypothalamus homogenates of conventional CerS6-deficient mice (CerS6^{Δ/Δ}) and conditional CerS6^{ΔNkx2.1} mice in the revised version of the manuscript. This analysis showed the absence of CerS6 protein in CerS6^{Δ/Δ} mice and a strong reduction of CerS6 protein levels in the hypothalamus of CerS6^{ΔNkx2.1} mice as compared to respective control mice. The remarkable reduction of CerS6 protein in the hypothalamus of

CerS6^{ΔNkx2.1} mice nicely underlines the predominant expression of CerS6 in neurons of the hypothalamus as was also revealed by single-cell sequencing results (see Figure 1 of the revised version of the manuscript).

In Figures 2-5, the authors only superficially mention the possibility that CerS6 deletion could affect development. In Figure 2, they make the case that rodent length is unchanged and use this as proof that there are no developmental effects on the mice; while it may be true that there are no growth effects, body length does not account for the possibility of developmental issues with neural circuitry that can affect feeding and obesity regulation throughout/later in life. A discussion of the possible developmental caveats of the Nkx2.1 and Pomc Cre mice should be included in this manuscript.

As this study was not aimed at investigating mouse development, we have deleted the sentence on mouse development and did not relate body size to possible developmental deficits to avoid that the reader would draw any conclusion that are not sufficiently supported by our data. In addition, we have pointed out the potential caveats arising from constitutive Cre mice in the revised version of the manuscript.

The authors state for Figure 2e, and other following figures, that reduced adiposity is consistent with reduced circulating leptin levels. While it is possible that reduced adiposity may result in less leptin production, a reduction in leptin should result in increased eating behavior and increased obesity/adiposity. These statements need to be clarified in the text. Leptin tolerance tests are also highly suggested to determine if leptin signaling is altered. Figure 5 should include circulating leptin, and for these Pomc-Cre mediated knockout of CerS6 mice, a leptin tolerance test is critical to know if these neurons are functioning correctly.

We thank the reviewer for these valuable suggestions. We have now included a leptin tolerance test in the revised version of the manuscript, i.e., immunohistochemical analysis of leptin-stimulated pSTAT3 immunoreactivity in POMC neurons of CerS6^{ΔPOMC} mice. This analysis revealed that HFD-fed obese CerS6^{ΔPOMC} mice are more sensitive to IP-injected leptin with respect to neuronal

pSTAT3 signaling as compared to HFD-fed obese control mice. This data is now presented in Figure 6 of the revised version of the manuscript.

Figure 2 h-j and subsequent figures, "genotype" and "interaction" need to be defined in the figure legend so that the reader understands what the two p values represent.

We thank the reviewer for bringing this to our attention. We have included a thorough description of the statistical analysis in the Methods section of the revised version of the manuscript.

Extended Data Figure 5 shows little association of CerS6 transcript with Pomc transcript. However, the variant CerS6 transcript does associate more closely. Better images or quantification of the loss of CerS6 need to be shown. In addition, neurons from other nuclei within the hypothalamus that arise from Pomc expressing founder cells during development should also be quantified for CerS6 loss.

We agree with the reviewer's concern that the staining and imaging quality of the ISH experiments was initially limited. According to the reviewer's suggestion, we have intensely optimized our BaseScope protocol and have now included higher quality images for the ISH-based analysis of CerS6 expression in the hypothalamus of all different mouse models employed in this study. These experiments clearly show the reduction of full-length CerS1 or CerS6 and expression of the genetically modified transcripts in the respective knockout mouse models.

In the results section for Figure 6, the authors need to explain why Xbp1 splicing is important.

We thank the reviewer for bringing to our attention that we have not properly explained our rationale of analyzing Xbp1 expression. In the revised version of the manuscript, we have now explained why Xbp1 splicing is important.

It has been shown that leptin signaling is altered in Pomc neurons with

increasing glucosylceramides. Therefore, figure 6 would be further enhanced by adding glucosylceramide data.

We agree that analyzing glucosylceramide levels would further improve the manuscript. We have thus included hexosylceramide data (including glucosyl- and galaktosylceramides) of N43/5 cells treated with siRNA and palmitate in Figure 2 of the revised version of the manuscript (initially Figure 6). This experiment indeed revealed an interesting phenotype, with hexosylceramides accumulating in response to siCerS6 treatment that may have occurred due to a compensatory upregulation of the sphingolipid salvage pathway, as we also discuss now in the revised manuscript. To investigate whether a similar response would occur *in vivo*, we additionally analyzed hexosylceramide levels in hypothalamus homogenates of conventional CerS6-deficient mice fed a HFD. Yet, in the hypothalamus of mice, CerS6 deficiency did not alter hexosylceramide levels.

The mHypoE-N43/5 cells used for Figure 6 are derived from female mice. All of the data presented in this manuscript for *Agrp* and *Pomc* Cre animals are from male mice, and the SF1 Cre mediated deletion CerS6 led to sexually dimorphic results. The study would be dramatically enhanced using primary neurons taken from the knockout mice generated by the lab.

We thank the reviewer for this comment. We have included a section in the discussion of the revised version of the manuscript in which we refer to the karyotype of N43/5 cells and relate this to our mouse models. In addition, we have now included new data on the phenotype of female mice of all relevant genotypes analyzed during the study.

Figure 6 makes a good case for CerS6 and C16 ceramide being deleterious for hypothalamic neurons. However, it makes a better Figure 1 introducing why the authors studied mice with a CerS6 knockout rather than correlating the data as a possible mechanism.

We have reorganized the figures according to the reviewer's suggestion. Figure 6 of the initial version of the manuscript is now Figure 2 of the revised version of the manuscript. Now we present the neuronal phenotypes *in vitro* immediately

after showing our observation that CerS6 is expressed predominantly in neurons of the hypothalamus and that CerS6 expression and C16:0 ceramide content increase in the hypothalamus of obese mice. We agree that the manuscript flow is clearly improved by this suggested structural change.

This lab has already established the importance of mitochondrial C16 ceramides; showing it in hypothalamic neurons is important, but the study would be significantly enhanced by showing mitochondrial dysfunction *in vivo*, not *in vitro*.

We fully agree with the reviewer that the conclusion of the manuscript would be even stronger if the mitochondrial phenotype was shown *in vivo*. We therefore analyzed mitochondrial shape specifically in POMC neurons of CerS6^{ΔPOMC} mice in collaboration with the group of Prof. Dr. Tamas Horvath at Yale University. Here, we found that the changes in mitochondrial morphology in POMC neurons, which are associated with HFD feeding, were prevented by deletion of CerS6, similar to what we have observed in siCerS6-treated cultured N43/5 cells exposed to palmitate *in vitro*. These data are now available in the revised version of the manuscript.

Sphingolipid data (concentrations and species) needs to be done in the hypothalamus from the mouse lines generated. Ideally, sphingolipids need to be obtained from the specific nuclei in which the CerS6 knockout occurs. For the Nkx2.1 Cre knockout mice, sphingolipid data should come from various regions of the brain. This is necessary to show that the knockout mice are functioning as expected and that exogenous sources of C16 ceramides aren't incorporated into the various regions/hypothalamic nuclei.

We absolutely agree that it would be ideal to present sphingolipid data of the different conditional knockout mice throughout the manuscript. Accordingly, we have measured ceramides in the hypothalamus homogenates of CerS6^{ΔNkx2.1} and CerS1^{ΔNkx2.1} mice, in which the ceramide synthases are targeted in a large proportion of hypothalamic neurons, but we were not able to detect significant changes in C16:0 and C18:0 ceramides, respectively. This is most likely due to the fact that the ceramide profile of hypothalamus homogenates mainly reflects that of non-neuronal cells as we now show in Extended Data Fig. 1i of the

revised version of the manuscript, thus we could not detect the change of ceramides as occurring in neurons of the knockout mouse models in crude hypothalamus homogenates. In addition, as also explained above, due to the limited number of POMC/AgRP neurons in the ARC and SF-1 neurons in the VMH, validation of a reduction in C16:0 ceramide levels in the hypothalamus of the different neuron-specific CerS6 knockout mouse models (CerS6^{ΔAgRP}, CerS6^{ΔPOMC}, CerS6^{ΔSF-1}) in crude hypothalamus homogenates or microdissections of hypothalamic nuclei was not feasible.

Still, to estimate whether CerS6 deficiency would result in an additional reduction or compensatory increase of other ceramide species in the hypothalamus, we analyzed the ceramide, sphingomyelin, and hexosylceramide content in the hypothalamus of HFD-fed conventional CerS6-deficient mice. Here, we observed that CerS6 deficiency specifically reduces C16:0 ceramides levels in the hypothalamus of HFD-fed mice, along with increased C20:0 ceramides, but without a reduction in other ceramide subtypes. This data is now available in the revised version of the manuscript.

The ER stress and mitochondrial phenotype seen in Figure 6 should be shown in the hypothalamus or at least in primary cultured neurons from the knockout mice. In addition, ER/mitochondrial stress markers such as those used in Figure 1 should be assessed in the knockout mice.

As stated above, we have now included TEM imaging results showing the mitochondrial phenotype in CerS6^{ΔPOMC} mice *in vivo*.

While it is easy to assume ER/mitochondrial stress would lead to neuron dysfunction, there is no evidence in this study that that happens when C16 ceramides are elevated in neurons. Functional data should be included.

In vitro we show that increased C16:0 ceramide synthesis, as induced by palmitate exposure, triggers ER/mitochondrial stress and alters mitochondrial morphology, which is also observed in POMC neurons *in vivo*. By analyzing the ability of leptin to induce STAT3 phosphorylation in POMC neurons in HFD-fed control and CerS6^{ΔPOMC} mice, we show that the alterations in mitochondrial

morphology correlate with changes in neuronal leptin sensitivity. We hope that this adequately addresses the reviewer's concern.

Line 382, extend should be extent.

We have revised the text according to the reviewer's suggestion.

Reviewer #2 (Remarks to the Author):

In the manuscript by Hammerschmidt et al, the authors investigated the role of CerS6-dependent ceramide synthesis in the hypothalamus and its role in regulating insulin resistance in murine models of obesity. The relevance of this work is high, as it adds details to the possible mechanism through which ceramides could control appetite and regulate metabolism in nutritional studies, suggesting that CerS6-generated ceramides are the main regulators and that this enzyme could be a possible therapeutic target. The authors made an important effort in generating several models to test their hypothesis. However, I am not convinced that the main conclusion, that Cer 16:0 is the main molecular messenger in the described model, can be fully supported without having measured ceramides in most of the reported results. I believe the current version of this manuscript should be significantly improved by the authors and I hope the following comments might clarify my conclusions.

We thank the reviewer for acknowledging the high relevance of our work and the generally very positive assessment of our manuscript. We have made great efforts to analyze ceramides and other sphingolipid levels *in vivo* to support our hypothesis that CerS6-derived ceramides in the hypothalamus contribute to the deterioration of metabolic control in obesity. Although we did not succeed in detecting a decrease in ceramide levels in crude hypothalamus extracts of the conditional knockout mice, we hope that by including additional experiments we have still adequately addressed the reviewer's concern and hopefully further convinced him/her about the proposed model. Surely, we cannot fully exclude that alternative functions of CerS6 independent of ceramide synthesis contribute to the phenotypes. However, such functions have not yet been described for CerS6 in the literature.

More in detail:

- In the introduction, a description of the specificity of each Ceramide synthase should be provided, to help readers understanding what is the possible overlap between these enzymes in terms of generated products; for example, CerS5 and CerS6 can both generate C16 ceramides. This I think will be a very useful information to include.

We thank the reviewer for this comment and have included a section on this topic in the introduction.

- This work is based on the elimination of CerS6 activity to significantly decrease the level of C16 ceramides. However, the product and the other ceramides level, that could be affected by the ablation of CerS6, are never measured in the target cells or in the whole hypothalamus. I think these data would be an important confirmation to support the authors findings and claims. But without measuring them, how can the authors be sure that CerS5 for example is not compensating for the lack of CerS6 activity? Was the measurement attempted other than in the invitro example? Please explain.

Please also see our answers to the questions of reviewers 1 and 3. We absolutely agree that it would be ideal to present ceramide data of the different conditional knockout mice throughout the manuscript. Accordingly, we have measured ceramides in the hypothalamus homogenates of CerS6^{ΔNkx2.1} and CerS1^{ΔNkx2.1} mice, in which the ceramide synthases are targeted in a larger proportion of hypothalamic neurons, but we were not able to detect significant changes in C16:0 and C18:0 ceramides, respectively.

This is most likely due to the fact that the ceramide profile of hypothalamus homogenates mainly reflects that of non-neuronal cells as we now show in Extended Data Fig. 1i of the revised version of the manuscript, thus we could not detect the change of ceramides as occurring in neurons of the knockout

mouse models in crude hypothalamus homogenates, particularly in mouse models, where only a minor proportion of neurons in an area has been targeted.

In addition, as also explained above, due to the limited number of POMC/AgRP neurons in the ARC and SF-1 neurons in the VMH, validation of a reduction in C16:0 ceramide levels in the hypothalamus of the different neuron-specific CerS6 knockout mouse models (CerS6^{ΔAgRP}, CerS6^{ΔPOMC}, CerS6^{ΔSF-1}) in crude hypothalamus homogenates or microdissections of hypothalamic nuclei was not feasible.

Still, to estimate whether CerS6 deficiency would result in an additional reduction or compensatory increase of other ceramide species in the hypothalamus, we analyzed the ceramide, sphingomyelin, and hexosylceramide content in the hypothalamus of HFD-fed conventional CerS6-deficient mice. Here, we observed that CerS6 deficiency specifically reduces C16:0 ceramides levels in the hypothalamus of HFD-fed mice, along with increased C20:0 ceramides, but without a reduction in other ceramide subtypes.

These new data are provided in the revised version of the manuscript.

- What is the meaning of the first sentence of the Summary "Ceramides form a heterogeneous group of acyl chain length-specific sphingolipids..."? considering that almost all lipids are.

We apologize for this inaccuracy and have changed it in the revised version of the manuscript.

- Abbreviations are used in the Summary before explanation of their meaning, please amend.

We have changed it in the revised version of the manuscript according to the reviewer's suggestion.

- Introduction lines 79-83: please explain a bit more in detail, by introducing a couple of examples of the findings.

We have changed it in the revised version of the manuscript according to the reviewer's suggestion.

- Lines 84-85: ceramides are not only differing by their acyl chain lengths but also by saturation state and long chain base type. Please amend to give a full picture of their heterogeneity.

We have changed this description in the revised version of the manuscript according to the reviewer's suggestion.

- Line 97: please specify which inhibitor the authors are referring to.

We have specified the inhibitor used in the revised version of the manuscript.

- Lines 134-135: how was ATF4 selected? As it is the only gene mentioned here.

Due to a change in the structural outline of the manuscript, we have removed the Western blot analysis of ATF4 in the hypothalamus of HFD-fed mice.

- Lines 137-39: in which mouse model was this done?

We have revised this analysis and included data of single-cell sequencing results from wild type mice harmonized in HypoMap (Steuernagel et al., 2023, Nat. Metab.).

- I understand that CerS5 might have been excluded as a possible target due to previous analysis made by the authors and reported in literature. But its distribution in the hypothalamus is very similar to CerS6. Why ablation of CerS5 was not considered as a control here? For example, in the paragraph starting at line 198 the authors describe the inhibition of CerS1, which is a good control. But CerS5 could have been added as well.

We agree with the reviewer that knockout of CerS5 could have been added as a control, in particular since we found CerS5 mRNA expression to be increased in crude hypothalamus extracts of obese mice compared to that of lean controls. As we now also state in the revised version of the manuscript, several reasons have let us not to focus on CerS5. First, we have previously shown that conventional deletion of CerS6 but not CerS5 protects mice from diet-induced obesity and the development of metabolic diseases, suggesting that CerS5 has a less critical role in the regulation of metabolic homeostasis than CerS6. Second, in cultured hypothalamic neurons (N43/5 cells) we found that only CerS6 mRNA expression increases with palmitate exposure (Figure 2 of the revised version of the manuscript), indicating that the increase in CerS5 mRNA that we observed in the hypothalamus of obese mice is either due to other obesity-related, palmitate-independent factors or due to regulation in non-neuronal cells. As we now discuss in the revised version of the manuscript, this latter hypothesis is also based on our observation from the scRNA-Seq results that CerS5 is expressed across various cell types in the hypothalamus, while CerS6 is predominantly expressed in neurons. Thus, we sought to focus our study on CerS6. CerS1 was used as control as C18:0 ceramides were the most abundant ceramide species in crude hypothalamus extracts and also account for a significant portion of the total ceramide content in neuron-enriched cell fractions (Extended Data Fig 1i of the revised version of the manuscript). Thus, in particular, we wanted to investigate whether there is a difference between hypothalamic CerS6-derived C16:0 ceramides versus CerS1-derived C18:0 ceramides. We agree, that the analysis of CerS5 deficient mice would be interesting, but we hope that the reviewer agrees, that this would extend the scope of the already extensive study.

- Lines 242-266: the authors describe an interesting sex-specific effect. Was this observed/excluded in the experiments reported in the previous paragraphs? A more detailed discussion on sex dimorphism reported earlier in similar conditions would be useful to the Discussion section.

First, we have extended the phenotyping of mice of both sexes in the revised version of the manuscript as well as have discussed observed sexual dimorphic responses more extensively.

- Lines 345-350: a similar effect can be observed for C24:0 Ceramides, any comment on this? As CerS6 is also generating C14 ceramides, and these are showing a similar profile as C16, why the author excluded the possibility of C14 being relevant in this model?

We apologize for this inaccurate description, we agree that C14 ceramides might also contribute to the observed phenotypes and have clarified this in the revised version of the manuscript.

- Line 375-85: this is a possible conclusion, although CerS can also generate deoxy-ceramides that have been shown to induce mitochondrial fragmentation (Alecu et al, 2017 JLR). Was that possibility considered?

We thank the reviewer for this comment. We have now included lipidomic data of deoxy-ceramide levels in N43/5 cells treated with siCerS6 and palmitate in the revised version of the manuscript. However, deoxy-ceramide levels were very low in N43/5 cells as compared to the ceramide or sphingomyelin content.

- More in general: CerS directly generate dehydro-ceramides but the authors only mentioned ceramides in the manuscript. Was the possibility of dehydro forms being the main player in this model considered? Please comment on this, especially as the authors always refer to de novo synthesis of sphingolipids.

We have also included lipidomic data on dihydroceramide levels in N43/5 cells. As proof of concept, siCerS6 treatment decreased C16:0 dihydroceramide levels as well. Although it cannot be fully excluded, it has been shown by others (e.g., Chaurasia et al., 2019, Science) that dihydroceramides are rather unlikely to cause metabolic alterations, which led us hypothesize that the metabolic effects observed by deleting CerS6 result from alterations downstream of dihydroceramide production, i.e. ceramide synthesis.

- Lines 405-408: the link between infusion of C6 ceramides and effect onto C16 and 18 ceramides level is not clear. Please explain.

We hope to have better explained the link of short-chain ceramide analogs with endogenous ceramide species in the revised version of the manuscript. While it is known by now that naturally non-occurring short chain ceramide analogs do not resemble the physicochemical characteristics of endogenous ceramide species, it has been shown by the Ogretman group that ceramide analogs are readily degraded when applied to cells or tissues while the degradation products can be used for endogenous ceramide formation. Consistent with these observations, Contreras et al (2014) have seen that icv infusion of C6 ceramide leads to increased production of C16:0 but not C18:0 ceramide levels in the hypothalamus of mice.

- Lines 408-413: this is a possible explanation, although, without measuring all the ceramides, we cannot exclude that for example accumulation of longer species might play a role too.

We have included full panels of ceramides in N43/5 cells treated with siCerS6 and palmitate, as well as of conventional CerS6-deficient mice. However, as also described in our responses to reviewer 1, we have measured ceramides in the hypothalamus homogenates of CerS6^{ΔNkx2.1} and CerS1^{ΔNkx2.1} mice, in which the ceramide synthases are targeted in a large proportion of hypothalamic neurons, but we were not able to detect significant changes in any ceramide species. This is most likely due to the fact that the ceramide profile of hypothalamus homogenates mainly reflects that of non-neuronal cells as we now show in Extended Data Fig. 1i of the revised version of the manuscript, thus we could not detect the change of ceramides as occurring in neurons of the knockout mouse models in crude hypothalamus homogenates. In addition, as also explained above, due to the limited number of POMC/AgRP neurons in the ARC and SF-1 neurons in the VMH, validation of a reduction in C16:0 ceramide levels in the hypothalamus of the different neuron-specific CerS6 knockout mouse models (CerS6^{ΔAgRP}, CerS6^{ΔPOMC}, CerS6^{ΔSF-1}) in crude hypothalamus homogenates or microdissections of hypothalamic nuclei was not feasible.

Still, to estimate whether CerS6 deficiency would result in an additional reduction or compensatory increase of other ceramide species in the hypothalamus, we analyzed the ceramide, sphingomyelin, and hexosylceramide content in the hypothalamus of HFD-fed conventional CerS6-deficient mice. Here, we observed that CerS6 deficiency specifically reduces C16:0 ceramides levels in

the hypothalamus of HFD-fed mice, along with increased C20:0 ceramides, but without a reduction in other ceramide subtypes.

- Line 466: "...critical role of a CerS6-derived ceramide/ER stress axis..." it would be extremely interesting to estimate the fold change in ceramide levels required to trigger this response. Was this done?

We agree that this would be an interesting subject for investigation, yet we hope that the reviewer agrees that this is beyond the scope of the already presently extensive study.

- Line 469: what is the example of C6 ceramide infusion telling us? Is it not going against the existence of acyl chain-specific effect of your model? Please explain.

Please also see our answer to the question above. It has been shown that membrane permeable short chain ceramide analogs are degraded when applied to cells or tissues while the degradation products can be used for endogenous ceramide formation. Effects observed by treatment with such ceramide analogs can thus be partially attributed to secondary effects by endogenous ceramides. Still, we fully agree that using short chain ceramide analogs alone is highly insufficient to draw conclusions about the functional role of specific acyl chain length ceramide species. This is why we have targeted specific ceramide synthases in hypothalamic neurons, to block the production of particular ceramide species. Interestingly, Contreras 2014 et al have reported that C6 ceramide infusion leads to increased concentration of C16:0 but not C18:0 ceramide levels in the mediobasal hypothalamus, associated with increased ER stress and feeding-independent weight gain. Thus, these data support the notion that C16:0 ceramide production in the hypothalamus is involved in hypothalamic ER stress and systemic metabolic control, as we have shown by deleting CerS6 to specifically inhibit CerS6-dependent C16:0 ceramide formation.

- The lipidomic methodology is not described in the method section. I understand it was previously reported in another publication but most of the procedure should also be reported here to help readers.

We thank the reviewer for this comment. We have included thorough descriptions of the lipidomic methodology in the revised version of the manuscript.

Reviewer #3 (Remarks to the Author):

This manuscript by Hammerschmidt et al proposes new cell type-specific mechanisms through which ceramides, and their synthetic enzymes, regulate metabolism. The studies presented here use a nice combination of molecular genetics, behavior, physiology, and in vitro molecular studies to demonstrate the role of ceramide synthase 6 (CerS6) in regulating metabolic outcomes in diet-induced obese mice. The work presented here is novel, appropriate in scope for the journal, and will provide a nice resource for understanding the cell-selective effects of ceramides in the hypothalamus. Below are some comments that might help to clarify some of the findings from this work.

We thank the reviewer for the positive assessment of our work.

Major comments:

Diverging effects of cell type specific KOs – how do the authors explain the divergence of effects for the near-whole hypothalamus CerS6 KO (in the Nkx2.1-Cre line) versus the cell subtypes (in this case, where effects were shown – SF1 and POMC)? For example, there is a strong effect on insulin sensitivity in the SF1 cKO, but not in the Nk2.1 cKO. Additionally, the GTTs at 6 and 16 hr do not accord across the different cKO lines.

The reviewer has pointed out important points. First we have aligned all reported data to report both 6 hr and 16 hr GTTs, which reflect different physiological states. These data are provided in the revised version of the manuscript. We agree that there is divergence across the cell type specific models. First of all, we feel that demonstrating the physiologically relevant role of CerS6 derived sphingolipids in hypothalamic neurons in itself is novel and important information. Second, we hypothesize, that the Nkx2.1 models captures way

more heterogeneous groups of neurons, than the ones subsequently studied via specific targeting, i.e. SF-1, POMC, AgRP neurons. This remarkable heterogeneity has been recently revealed by our hypothalamic single cell atlas HypoMap (Steuernagel et al., Nat. Metab. 2022). Thus, we speculate that CerS6 in other neuronal subtypes may serve differential, and even opposing functions, which lead to the dissociation of some phenotypes between Nkx2.1 targeted mice versus more specific models.

Magnitude of effect of metabolic state on CerS6 expression – CerS6 in the hypothalamus is significantly, but only modestly, increased in expression at the level of mRNA in db/db, HFD (although the dispersion of CerS6 protein expression is much broader); whereas, some of the metabolic phenotypes are more robust. What would account for this? Is this simply due to cell-selective changes of CerS6 expression across different hypothalamic subtypes? Or do the authors think that there may also exist other adaptations, such as post-translational modifications of CerS6 that could account for the observed phenotypes?

Again, the reviewer raises an interesting topic. We are currently investigating the cell type specific gene expression data of hypothalamic neurons during HFD via sn sequencing. However, these extensive studies are beyond the scope of the current manuscript. Also, posttranslational modifications of CerS6 represent an interesting possibility. Indeed, phosphorylation sites have been identified in CerS6, nevertheless to our knowledge no previous study has addressed the functional consequences of these phosphorylation events.

Minor comments:

Fig. 1 – can the authors comment further on the (marked) increase in CerS5, relative to CerS6 (or all other CerS enzymes in HFD, for that matter)? Might their hypothesized differential control of subcellular ceramide pools lead to distinct metabolic outcomes (as suggested in Hammerschmidt et al., Cell 2019)? In other words, are CerS5/6 gene expression changes hypothesized to act in an oppositional fashion? Also, does CerS5 expression correlate with body weight, as well?

As also pointed out in response to reviewer's 1 comments, we agree it will be interesting to study CerS5 more specifically in the future. However, given our previous findings that whole body CerS5 deficiency has very little effect on energy homeostasis and glucose metabolism in obesity, we did not further investigate its hypothalamus-specific role. Moreover, our single cell analysis now newly provided and refined in the revised version of the manuscript revealed that CerS6 exhibits a much more neuron-restricted expression pattern compared to the other ceramide synthases, further supporting to focus on CerS6 analyses in neuronal function. Still, we have extended the discussion section on CerS5 explaining our rationale to study CerS6 but CerS5 in greater detail.

Fig. 2 – the CerS6 cKO in Nkx2.1-Cre leads to a small, but significant decrease (relative to controls) in adiposity and body weight, when the mice are placed on HFD. However, there is no observed change in energy intake or expenditure. What do the authors think is happening here?

The reviewer is absolutely right that a change in body weight is hard to explain when food intake and energy expenditure remain similar between the groups. We have thus re-analyzed the energy expenditure data of CerS6^{ΔNkx2.1} mice by analysis of covariance using body-weight as a covariant. Indeed, this analysis showed that CerS6^{ΔNkx2.1} mice show a clear trend toward increased energy expenditure that is not only driven by the differences in the mouse body mass, explaining that the reduction in body weight of CerS6^{ΔNkx2.1} mice is likely due to the genotype-driven increase of energy expenditure. These data are now provided in Figure 3 of the revised version of the manuscript.

Fig. S1 – addition of dot plots in the scRNA-seq analysis would be useful to compare expression of the CerS isoforms across populations (i.e. SF1, Agrp, Pomc).

We thank the reviewer for the suggestion to present dot plots for better illustration of the expression of the different CerSs in the hypothalamus of mice according to the single cell sequencing (scRNA-seq) results. We have included dot plots of the scRNA-seq results in Figure 1 and Extended Data Figure 1 of

the revised version of the manuscript. This analysis nicely illustrates that CerS6 is expressed predominantly in neurons of the hypothalamus compared to other major cell types and that CerS6 is expressed in all three populations of hypothalamic neurons (i.e., SF1, AgRP, POMC), while at the same time showing the CerS distribution across different cells of the hypothalamus.

REVIEWERS' COMMENTS

Reviewer #1 (Remarks to the Author):

The authors have addressed all of my concerns.

Reviewer #3 (Remarks to the Author):

The authors have sufficiently addressed all of my concerns. The new analyses are a nice addition to the paper.

Reviewer #4 (Remarks to the Author):

The manuscript by Hammerschmidt et al. describes the role of C16:0- dependent Ceramide synthase CerS6 in the hypothalamus and its regulation in metabolic homeostasis. The authors used several mouse models as well as hypothalamic murine neurons to show that palmitate leads to increased levels of CerS6, which in turn is connected with ER/mitochondrial stress. The authors conclude that CerS6-dependent ceramide synthesis might also be a therapeutic target against obesity and diabetes. This work is of novelty and gives important functional insights on C16:0 induced ceramide synthesis and its correlation to metabolism.

The authors of the revised manuscript addressed and answered all concerns in a detailed and critical manner.

In more detail:

The authors added missing information regarding mouse models and content (e.g.: description of different Ceramide synthases, unclear statements/expressions, missing explanation for abbreviations, sex-specific effects), but also methodology applied.

Reviewer #2 stated previously, that CerS can also generate dehydro-ceramides and this should be considered in the manuscript. The authors addressed this concern and included the missing data in the revised manuscript.

In the previous round of the reviewing process the concern was raised, that the authors could have added CerS5 as additional control. This would have been of interest, but I also agree with the authors that having additional mouse models would have extended the

scope of this work. In addition, the authors explained very well, that due to previous results obtained the focus was set on CerS6, and not on CerS5.

Concerning the lipidomics methodology applied throughout the study: The methodology is sound and additionally has been published previously (see reference:84-87 in the manuscript).

In summary the manuscript has been revised to a great extent and should be accepted for publication.

Specifically, we have addressed the reviewers' concerns as follows:

Reviewer #1 (Remarks to the Author):

The authors have addressed all of my concerns.

Reviewer #3 (Remarks to the Author):

The authors have sufficiently addressed all of my concerns. The new analyses are a nice addition to the paper.

Reviewer #4 (Remarks to the Author):

The manuscript by Hammerschmidt et al. describes the role of C16:0- dependent Ceramide synthase CerS6 in the hypothalamus and its regulation in metabolic homeostasis. The authors used several mouse models as well as hypothalamic murine neurons to show that palmitate leads to increased levels of CerS6, which in turn is connected with ER/mitochondrial stress. The authors conclude that CerS6-dependent ceramide synthesis might also be a therapeutic target against obesity and diabetes. This work is of novelty and gives important functional insights on C16:0 induced ceramide synthesis and its correlation to metabolism.

The authors of the revised manuscript addressed and answered all concerns in a detailed and critical manner.

In more detail:

The authors added missing information regarding mouse models and content (e.g.: description of different Ceramide synthases, unclear statements/expressions, missing explanation for abbreviations, sex-specific effects), but also methodology applied.

Reviewer #2 stated previously, that CerS can also generate dehydro-ceramides and this should be considered in the manuscript. The authors addressed this concern and included the missing data in the revised manuscript.

In the previous round of the reviewing process the concern was raised, that the authors could have added CerS5 as additional control. This would have been of interest, but I also agree with the authors that having additional mouse models would have extended the scope of this work. In addition, the authors explained very well, that due to previous results obtained the focus was set on CerS6, and not on CerS5.

Concerning the lipidomics methodology applied throughout the study: The methodology is sound and additionally has been published previously (see reference:84-87 in the manuscript).

In summary the manuscript has been revised to a great extent and should be accepted for publication.

We thank the reviewers for their kind and positive assessment of our revised manuscript.